# Deep mutational scanning and machine learning reveal structural and molecular rules governing allosteric hotspots in homologous proteins

Megan Leander[1†], Zhuang Liu[2†], Qiang Cui[2,3]*, Srivatsan Raman[1,4,5]*

[1]Department of Biochemistry, University of Wisconsin-Madison, Madison, United States; [2]Department of Physics, Boston University, Boston, United States; [3]Department of Chemistry, Boston University, Boston, United States; [4]Department of Bacteriology, University of Wisconsin-Madison, Madison, United States; [5]Department of Chemical and Biological Engineering, University of Wisconsin-Madison, Madison, United States

**\*For correspondence:**
qiangcui@bu.edu (QC);
sraman4@wisc.edu (SR)

[†]These authors contributed equally to this work

**Abstract** A fundamental question in protein science is where allosteric hotspots – residues critical for allosteric signaling – are located, and what properties differentiate them. We carried out deep mutational scanning (DMS) of four homologous bacterial allosteric transcription factors (aTFs) to identify hotspots and built a machine learning model with this data to glean the structural and molecular properties of allosteric hotspots. We found hotspots to be distributed protein-wide rather than being restricted to 'pathways' linking allosteric and active sites as is commonly assumed. Despite structural homology, the location of hotspots was not superimposable across the aTFs. However, common signatures emerged when comparing hotspots coincident with long-range inter-actions, suggesting that the allosteric mechanism is conserved among the homologs despite differences in molecular details. Machine learning with our large DMS datasets revealed global structural and dynamic properties to be a strong predictor of whether a residue is a hotspot than local and physicochemical properties. Furthermore, a model trained on one protein can predict hotspots in a homolog. In summary, the overall allosteric mechanism is embedded in the structural fold of the aTF family, but the finer, molecular details are sequence-specific.

## Editor's evaluation

This article seeks to address a key question in protein biophysics: are the amino acid positions involved in allosteric mechanisms conserved across homologs of a protein family? Or do these mechanisms involve distinct amino acid patterns that vary amongst homologs? To address this question, the authors follow an innovative multidisciplinary approach that combines deep mutational scanning with machine learning; the findings of this study will be highly relevant to protein engineers and biophysicists.

## Introduction

Allostery is a fundamental regulatory mechanism governing proteins involved in diverse biological functions (*Changeux and Edelstein, 2005*). It is a fascinating property of proteins where perturbation at one site of a protein elicits a response at a distant site but one whose molecular principles remain poorly understood (*Wodak et al., 2019*). Though allosteric proteins may employ diverse structural

mechanisms to propagate the perturbation, all allosteric proteins obey a simple thermodynamic principle that binding an effector ligand stabilizes the active state over the inactive state and removing the effector ligand reverses this effect (*Changeux, 2012*; *Cui and Karplus, 2008*; *Marzen et al., 2013*; *Hilser et al., 2012*). We need to investigate the molecular nature of allostery at the residue level to understand how diverse structural mechanisms are bound by the same thermodynamic principle; that is, do common underlying molecular 'rules' of allostery exist? To answer this question, we need to identify the allosteric 'hotspots' or residues critical for allosteric signaling. However, the location of allosteric hotspots cannot be gleaned from structure alone. Our recent deep mutational scanning (DMS) analysis of bacterial transcription factor, TetR, revealed that allosteric hotspots are distributed throughout the protein with no apparent direct structural link to either the allosteric or the active site (*Leander et al., 2020*). In contrast, the commonly held view is that hotspot residues tend to fall along well-defined pathways linking both sites (*Ota and Agard, 2005*; *Süel et al., 2003*; *Strickland et al., 2008*; *Reynolds et al., 2011*; *Amor et al., 2016*). In recent years, targeting allosteric rather than active site has also emerged as an attractive therapeutic strategy, especially for drug targets that implicate ubiquitous molecules, such as ATP, as the substrate (*Nussinov and Tsai, 2013*; *Abdel-Magid, 2015*). Therefore, from both fundamental and biomedical application points of view, it is important to develop methodologies to systematically identify allosteric hotspot residues and to understand the molecular nature of these residues.

There are no well-established methods for determining the location of allosteric hotspots in a protein. Current experimental approaches impute allosteric hotspots from residue connectivity in crystal structures (*del Sol et al., 2006*) or changes in NMR chemical shifts or dynamics (*Tzeng and Kalodimos, 2009*; *Lisi et al., 2016*; *Guo and Zhou, 2016*). These approaches at best identify only a subset of hotspots, but may also misidentify hotspots simply because they lie in-between allosteric and active sites or show local motion. Other metrics to assign the importance of a residue such as shortest path length (*Vanwart et al., 2012*; *Sethi et al., 2009*) or density of connections (*Wang et al., 2020*) are only tangentially related to allostery. Computational approaches have limitations too. Sequence co-evolution patterns reveal statistically linked residue pairs (*Süel et al., 2003*; *Reynolds et al., 2011*), but face ambiguity regarding the origin of co-evolution, which can be driven by folding stability rather than allostery. Molecular dynamics simulations (*Cui and Karplus, 2008*; *Guo and Zhou, 2016*; *Papaleo et al., 2016*) in combination with analysis such as community network analysis (*Sethi et al., 2009*; *Rivalta and Batista, 2021*; *Nierzwicki et al., 2021*) or Markov state models (*Kuzmanic et al., 2020*) has been used to identify allosteric hotspots or cryptic allosteric sites, but these approaches are often not comprehensive. In recent years, DMS has emerged as a powerful tool to understand protein function by measuring the impact of mutational perturbations using high-throughput experiments (*Fowler et al., 2010*; *Fowler and Fields, 2014*; *Sarkisyan et al., 2016*; *Flynn et al., 2020*; *Starr et al., 2020*; *Huss et al., 2021*). DMS is particularly useful to study a systemic property like allostery because it permits an unbiased examination of every residue of a protein without a priori assumptions about its functional role (*Leander et al., 2020*; *Jones et al., 2019*; *Tack et al., 2020*; *Faure et al., 2022*; *McCormick et al., 2021*). Combining DMS with statistical tools allows us to recognize complex underlying patterns describing the molecular rules of allostery.

In this study, we used DMS to identify allosteric hotspots by systematically dissecting the functional contribution of each residue to allosteric signaling. Using this approach, we compared hotspots across four distant homologs in the TetR-like family of allosteric transcription factors (aTFs). We found that the location of allosteric hotspots is unique to each homolog despite similarities in allosteric signaling within this family. However, a common pattern emerges when comparing hotspots clustered around residues participating in long-range interactions (LRIs) suggesting that non-bonded, LRIs play a defining role in the transmission of signal between allosteric and active sites. We leveraged the DMS data to train a machine learning model (a feedforward neural network [NN] integrated with a GA for feature selection) using a broad set of local and global properties of the hotspots for classifying whether a residue is a hotspot. By analyzing the performance of different NN models and identifying features that dictate the accuracy of classification, we gained insights into factors that are likely essential to allostery. Finally, we explore the transferability of the NN model among homologous proteins. This helps elucidate to what degree the mechanism of allostery is conserved among proteins in the same family and the information content of models required to make a meaningful prediction of

allosteric hotspots. Our study lays the foundation for combining DMS and machine learning to infer molecular mechanism of allostery within a protein family.

## Results and discussion

### Identifying allosteric hotspots across homologs

aTF is an ideal model system because of its simple one-component signal transduction mechanism that can be converted into a reporter-based high-throughput screen to measure allosteric activity (*Leander et al., 2020*; *Nishikawa et al., 2021*). We chose to study the TetR family of transcription regulators because they are a large and remarkably diverse family of proteins found in almost every bacterial host with diverse ligand and DNA specificities (*Cuthbertson and Nodwell, 2013*). As a result, the allosteric mechanisms of these proteins have evolved under different selection pressures exerted by their environments. Despite their diversity, all TetR family proteins (>100 in PDB) share a similar protein structure which suggests their structure is versatile and robust to preserve allostery while accommodating diverse sequences (*Cuthbertson and Nodwell, 2013*; *Fukami-Kobayashi et al., 2003*). Therefore, the TetR family serves as a good model system to investigate structural properties of allostery common within the family while minimizing sequence-dependent effects. We chose four aTFs – TetR, TtgR, MphR, and RolR – with high structural similarity (between 1 and 3 Å Cα root mean squared distance [RMSD]) but low sequence identity (between 14% and 19%, *Supplementary file 1*).

To probe the functional impact of mutations on aTFs, we developed a high-throughput pooled screen in *Escherichia coli* where the activity of mutants can be measured by the expression level of GFP regulated by an aTF-regulated promoter. Allosteric activity was quantified as the fold induction ratio of GFP expression with and without the inducer. Fold induction of wild-type aTFs was TetR: 49-fold (ligand: anhydrotetracycline [aTC]), TtgR: 25-fold (ligand: naringenin [Nar]), MphR: 100-fold (ligand: erythromycin [Ery]), and RolR: 15-fold (ligand: resorcinol [Res]). We mutated each aTF using commercially available chip oligonucleotides to encode a comprehensive library of point mutants by single-site saturation mutagenesis of each residue (~200 residues per aTF × 19 mutants/residue = 3800 mutants per aTF, *Figure 1—figure supplement 1*). We designate aTF mutations that constitutively lock the protein in an inactive allosteric state as 'dead variants'. This may occur because the mutation stabilizes the inactive state by increasing the thermodynamic gap between inactive and active states. The dead variants are well-folded proteins that bind to DNA and repress transcription but cannot be induced with the ligand. From each aTF library, we enriched dead variants by sorting low GFP cells after incubation with their corresponding ligand (*Figure 1A*; *Figure 1—figure supplement 1*). The sorted populations were deep sequenced in triplicate to identify the allosterically dead variants (*Figure 1—figure supplement 2*; *Supplementary file 2*). In other words, we use cell sorting as a binary classifier; that is, does the mutation disrupt allostery or not. We capture the effect size on individual residues, not individual mutations, by counting the number of dead mutations at a residue position. This is an important consideration because it safeguards us from minor inconsistencies that inevitably arise from cell sorting.

Next, we wanted to establish criteria to designate a residue as an allosteric hotspot. The importance of a residue for allosteric signaling is proportional to the number of dead variants at that position. For example, a residue with 15 dead variants is more important than one with five dead variants. We cannot choose an arbitrary threshold for the number of dead variants as this threshold may change for each aTF. Therefore, we created a simple scoring system where each residue was given a score based on the number of dead variants at that position and the confidence a variant is fully dead. The latter criterion captures variants that show weak allosteric activity. A higher positional score indicates the higher importance of a residue in allosteric signaling. We designated residues falling in the highest quartile (top 25% scoring residues) in the interquartile distribution of scores as allosteric hotspots for each aTF. The spread of residue scores varied between aTFs. The highest quartile was well separated for TetR, TtgR, and MphR, and less so for RolR, giving us higher confidence in the assignment of hotspots in the former groups (*Figure 1—figure supplement 3*). We note that the lower fold induction (dynamic range) of RolR makes it particularly challenging to separate the dead variants from the rest. We designated 53, 51, 48, and 57 residues as hotspots in TetR, TtgR, MphR, and RolR, respectively. To assess the robustness of our classification of hotspots, we determined the number of

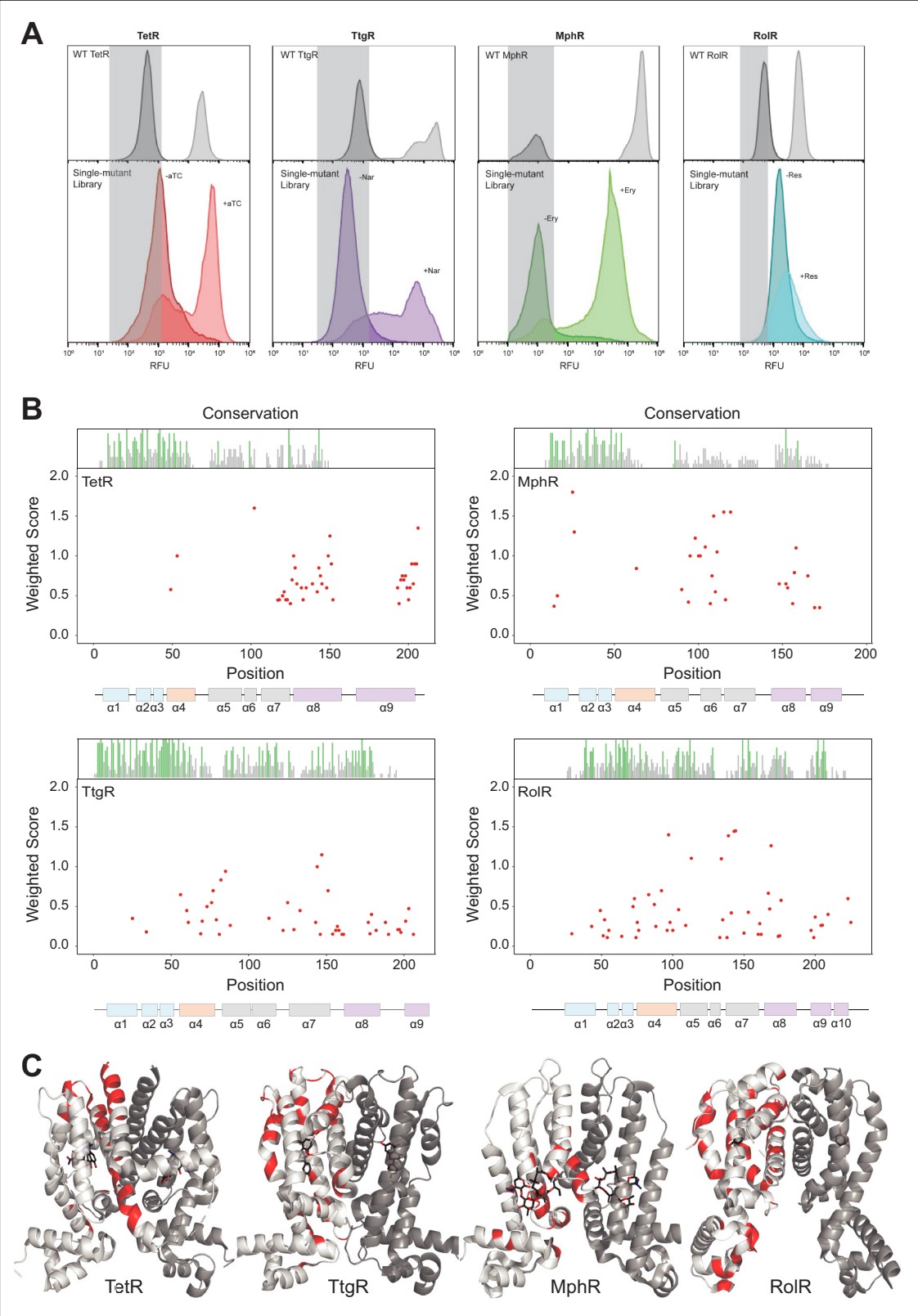

**Figure 1.** Allosteric hotspots in four bacterial allosteric transcription factors (aTFs) identified using deep mutational scanning (DMS). (**A**) Nonfluorescent cells in the TetR, TtgR, MphR, and RolR single-mutant library were sorted (gray bar) in the presence (light shade) and absence (dark shade) of 1 μM anhydrotetracycline (aTC), 500 μM naringenin (Nar), 1 mM erythromycin (Ery), and 7.5 mM resorcinol (Res), respectively, and sequenced to identify dead variants. Sorting gates were defined by the wild-type uninduced population for each homolog. (**B**) Allosteric hotspots (red points) for each aTF is

*Figure 1 continued*

shown with residue numbers along x axis and a weighted score along y axis based on the number of dead mutations at a residue position. Secondary structures of the aTFs are illustrated below and colored according to regions (blue: DBD, orange: hinge helix connecting LBD and DBD, gray: LBD and purple: dimer interface). Residue conservation is shown and colored by conserved residues (green), not conserved (gray) and conserved overlapping with hotspot (red). (**C**) Allosteric hotspots mapped on to the structure of TetR, TtgR, MphR, and RolR (ligand-contacting residues excluded).

The online version of this article includes the following figure supplement(s) for figure 1:

**Figure supplement 1.** Experimental scheme for deep mutational scanning.

**Figure supplement 2.** A detailed summary of all single-mutant phenotypes for every position within the proteins.

**Figure supplement 3.** Histograms of weighted scores and thresholds for identifying hotspots.

**Figure supplement 4.** Correlation of weighted scores between a ×5 or ×10 read count threshold.

**Figure supplement 5.** Distribution of allosteric hotspots in TetR homologs.

**Figure supplement 6.** Conservation of allosteric hotspots.

**Figure supplement 7.** Comparison of experimental hotspots with predictions made by the Ohm server.

hotspots at two different sequencing thresholds – ×5 and ×10. At ×5 and ×10, the number of hotspots is – TetR: 53, 51; TtgR: 51, 51; MphR: 48, 48, and RolR: 57, 60, respectively (*Figure 1—figure supplement 4*). Changing the threshold has a modest impact on the overall number of hotspots and the regions of functional importance are consistent at both thresholds. After excluding ligand-contacting residues from consideration, as mutations at these residues appeared dead likely due to loss of ligand affinity, we were left with 41, 43, 29, and 51 hotspots in TetR, TtgR, MphR, and RolR, respectively. We note that changing the read threshold does not change the identity of the hotspots falling in the top quartile indicating the robustness of our conclusions.

The location of hotspots on the structure was unique to each aTF despite their structural homology. Hotspots of TetR and MphR were concentrated in the C-terminal half of the protein (*Figure 1B*), whereas those of TtgR and RolR were distributed across the entire structure. This result exposes a key difference between understanding the allosteric mechanism at the level of protein structure vs. individual residues. The structural mechanism of allostery may be specified once a protein fold is specified. However, the residues involved in 'executing' the structural mechanism may be unique to different sequences folding into that structure. In other words, nature has created degenerate molecular pathways to transmit the allosteric signal within the same protein structure.

To understand the common structural mechanism in the family, we classified the hotspots based on their secondary structure location (*Figure 1B*) into α helices 1 through 9 (or 10): DNA-binding domain (DBD = α1, -2, -3), the ligand-binding domain (LBD = α5, -6, -7), the alpha helix connecting DBD and LBD (α4), and the dimer interface (α8, -9 for TetR, TtgR and MphR and α8, -9, -10 for RolR). Three structural similarities emerged in the location of hotspots across aTFs. First, a high fraction of hotspots, relative to the segment's length, were at the dimer interface (*Figure 1—figure supplement 5*). This suggests that allosteric signaling through the dimer interface is likely a conserved mechanism in TetR family. Second, a high fraction of hotspots was on α4 suggesting α4 acts as a mechanical link that transmits allosteric signals from the LBD to the DBD. Third, very few hotspots were in the DBD compared to other regions. This suggests the DBD is a standalone domain whose interaction with DNA is controlled by allosteric forces originating outside the DBD. An evolutionary perspective strengthens this hypothesis. The evolution of aTFs has occurred through a series of gene duplication events resulting in mixing and matching LBDs and DBDs (*Pougach et al., 2014*; *Yuan et al., 2022*). Thus, the DBDs likely exist as standalone domains that respond to large thermodynamic changes (e.g., inducer binding). Taken together, these observations show that although the hotspots are not superimposable across aTFs, the TetR family likely shares a conserved structural mechanism where the allosteric signal travels from the LBD through the dimer interface and α4 to the DBD, while the DBD itself acts as an internally rigid module that docks on DNA. Detailed biophysical or molecular dynamics characterization will be required to further validate our conclusions (*Gandhi et al., 2008*).

Since residues important for function (e.g., binding, catalysis, etc.) tend to be conserved in sequence, we assessed if allosteric hotspots too are conserved. We compared hotspots to close sequence homologs (>50% sequence identity) and did not find statistically higher sequence conservation in hotspots over non-hotspots (*Figure 1—figure supplement 6*). This reinforces our earlier conclusion that though the structural mechanism of allosteric signaling may be conserved within this family, the residues

participating in signal transduction may be specific for each aTF. In other words, allosteric sites are not necessarily conserved, though allostery itself may be conserved. We also compared the experimental hotspots with predictions made by the Ohm webserver (*Wang et al., 2020*). The Ohm webserver is an efficient computational tool that analyzes the propagation of structural perturbation in proteins to identify allostery network and hotspot residues. The overlap between predictions and experiments is modest and involves mostly DBD residues while the experimental hotspots are distributed across the protein (*Figure 1—figure supplement 7*). This highlights the limitation of focusing on the mechanistic model that involves propagation of conformational distortions.

## LRIs reveal similarities in allosteric mechanism

We investigated what underlying property of protein structure might explain the preference of hotspots for certain sites. We considered the defining characteristic of allostery, that is, cooperative action between spatially distant residues. Cooperative action occurs through molecular forces transmitted between bonded and non-bonded interactions. Forces transmitted through bonded interactions tend to dissipate over short distances. However, non-bonded interactions, particularly between residues farther in primary sequence, likely facilitate transmission of force over longer distances (*Miyazawa and Jernigan, 1996*). Therefore, we examined the location of allosteric hotspots with respect to residues involved in non-bonded LRI.

We generated contact maps of residue-residue interactions and selected LRIs as residues separated by 10 or more positions in sequence but within 8 Å in Cα- Cα distance. For each aTF, we then compared LRI residues and allosteric hotspots. In all four aTFs, hotspots constituted a higher fraction of LRIs than non-hotspots (*Figure 2—figure supplement 1*; p=0.07). Hotspots in TetR and MphR are especially enriched in LRIs – 28% hotspots vs. 17% non-hotspots. This gap is smaller in TtgR and RolR – 23% hotspots and 20% non-hotspots. It is worth noting that in addition to allostery, LRIs play an important role in protein folding and stability. Thus, it is not surprising that only a fraction of LRIs overlaps with allosteric hotspots. But this subset of LRIs may offer insight into the mechanism of allostery in the TetR family. Therefore, we grouped the LRIs using standard, unsupervised k-means clustering on the contact maps. The optimum number of clusters for each aTF was determined using a standard Elbow method that iteratively calculates the variance within clusters for different numbers of clusters. The optimal number of clusters is the point yielding diminishing returns (higher variance within a cluster) that is not worth the cost of adding new clusters and was found to be 10 clusters independently for each aTF (*Figure 2—figure supplement 2*).

The 10 clusters of LRIs represented distinct local regions of the protein (*Figure 2A*). To evaluate the relative importance of different signaling pathways, we ranked the LRI clusters based on the fraction of unique hotspots out of all residues in that cluster (*Supplementary file 3*). The dimer interface emerged as the first-ranked cluster in TetR, MphR, and RolR (*Figure 2B*), suggesting that the dimer interface is a dominant signaling pathway. The first-ranked cluster of TtgR was not the dimer interface which is consistent with far fewer hotspots found at the dimer interface of TtgR (*Figures 2 and 1C*). The second tier of cluster rankings contained regions between helices α4, -5, -6, and -7 of the LBD and the linker helix between LBD and DBD. These clusters likely represent allosteric forces emanating from the LBD upon ligand binding. No cluster stands out as dominant within this tier. LRI clusters within the DBD were ranked near or at the bottom of the rankings for all homologs. LRIs have long been known to play a key role in protein folding and stability. Our results show that LRIs are also critical for the propagation of allosteric signals. These results also show that though allosteric hotspots may not be superimposable across distant homologs, local clusters of LRIs share similar patterns between homologs. As homologs get closer in sequence, regional similarities in allosteric signaling may give way to the superimposability of individual hotspots.

## Physiochemical properties of dead mutations

We investigated if mutations to certain amino acids were enriched among dead variants over non-dead variants. We computed the percentage of each amino acid among mutations that were dead vs. not-dead from all four aTF datasets (~12,000 mutations). Aromatic amino acids (Phe, Trp, and Tyr) were enriched among dead variants (20%) over the not-dead group (15%) (*Figure 3A*). Mutations to proline were also enriched, albeit to a lesser degree, among the dead variants (5%) vs. the not-dead

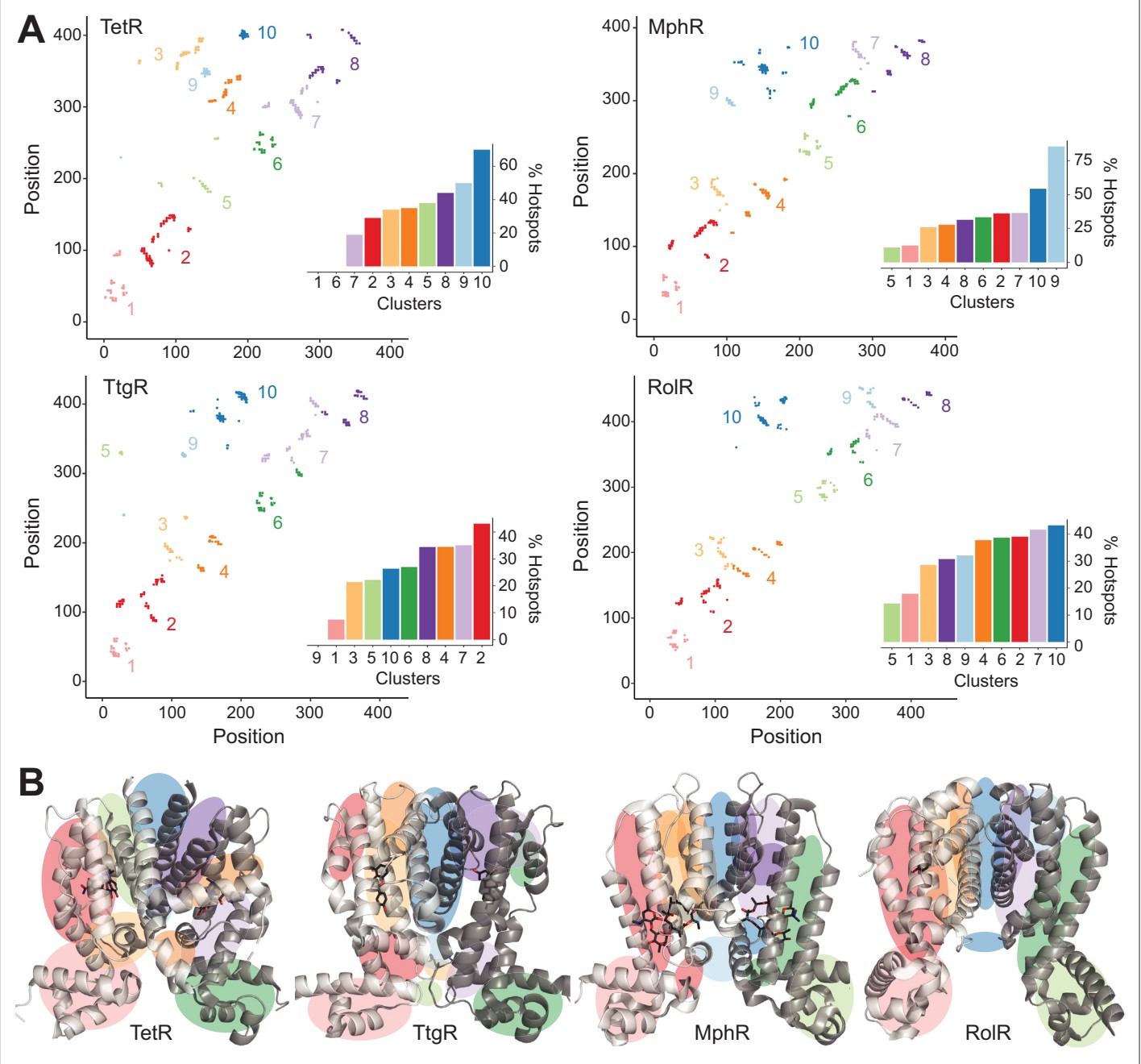

**Figure 2.** Hotspots enriched among long-range interactions (LRIs). (**A**) Residue-residue contact map showing LRIs within each homolog. The LRIs are grouped by color, following standard k-means clustering, representing different regions of the protein. Inset shows ranking of LRI clusters based on the percentage of unique hotspots within each cluster. (**B**) The general location of each LRI cluster on the protein structure (color scheme same as panel A).

The online version of this article includes the following figure supplement(s) for figure 2:

**Figure supplement 1.** Hotspot interactions are more likely to be long range than those of non-hotspots.

**Figure supplement 2.** Elbow method to determine the optimal number of clusters.

group (3.8%). These trends change slightly at the protein level, for example, the branched nonpolar leucine is the most enriched mutation in dead variants of TetR (*Figure 3—figure supplement 1*).

Next, we compared differences between both groups in six common physicochemical properties of amino acids. We did not observe any statistically significant differences in hydrophilicity, hydrophobicity, and polarity between both groups (*Figure 3B*). However, we observed statistically significant

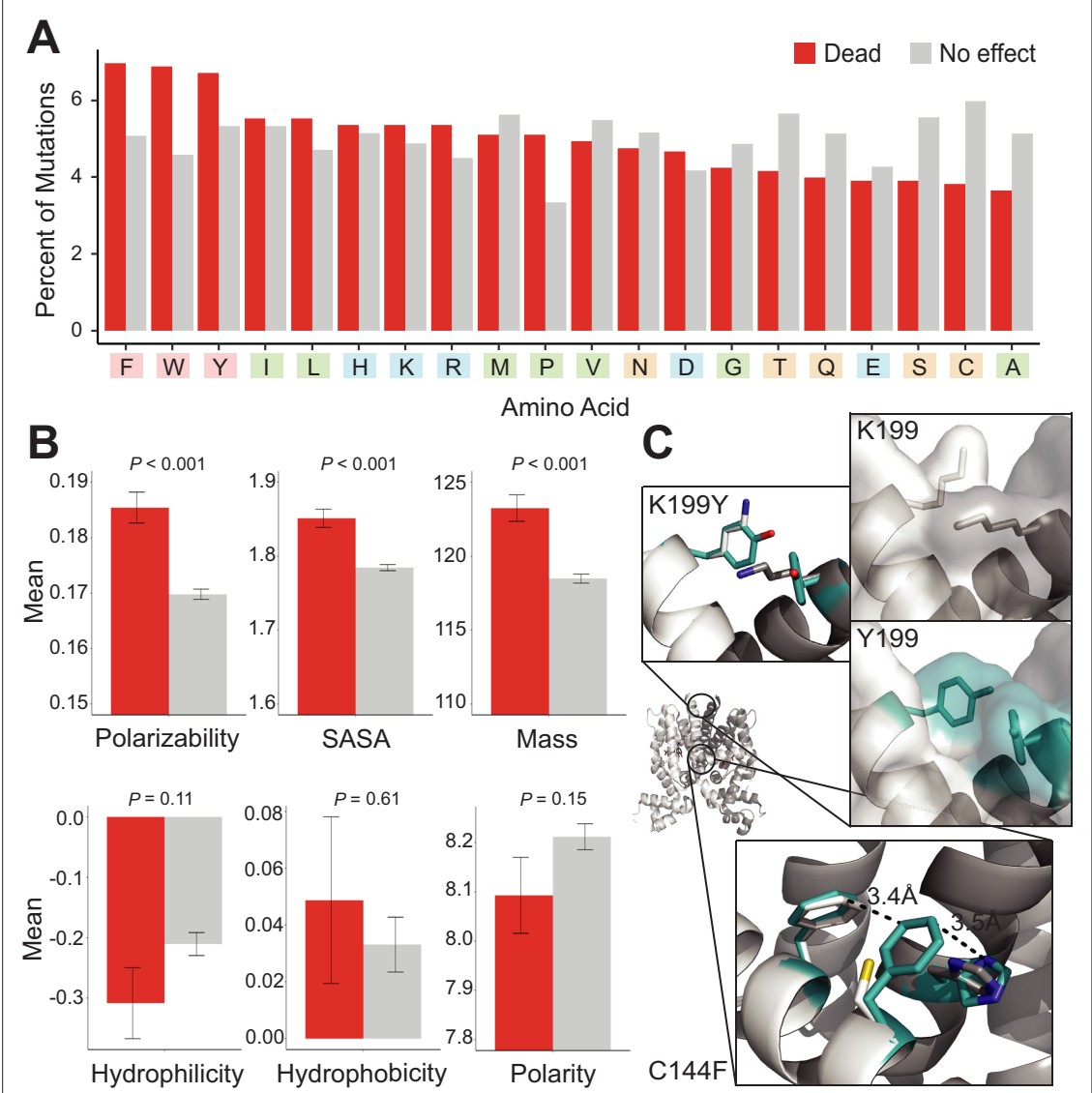

**Figure 3.** Mutational preferences and physicochemical properties of dead variants. (**A**) Percentage of mutations (final mutated state) among dead (red) and not-dead (gray) variants from deep mutational scanning (DMS) data for all four homologs combined. (**B**) Comparison of physicochemical properties – polarizability, solvent-accessible surface area (SASA), mass, hydrophilicity, hydrophobicity, and polarity – between dead (red) and not-dead variants (gray). Average values aggregated over all four DMS datasets shown. Data represented as mean ± SEM. (**C**) Structural models of the K199Y (top) and C144F (bottom) mutations in TetR. Residues in the mutant structures are colored teal and the two monomers are colored white and dark gray.

The online version of this article includes the following figure supplement(s) for figure 3:

**Figure supplement 1.** Enrichment of mutations in allosterically dead or no effect variants.

differences in polarizability, solvent-accessible surface area (SASA), and mass (*Figure 3B*). This is consistent with aromatic residues having larger mass and SASA, and greater polarizability due to the π-electron cloud. Although aromatic residues are also hydrophobic, hydrophobicity itself is not a differentiator between dead vs. not-dead groups. The enrichment of aromatic amino acids, and to a lesser extent strongly aliphatic amino acids (Ile and Leu) among the dead variants, hints at a relationship between residue-residue interaction energy and allosteric signaling. These amino acids (Trp, Phe, Tyr, Ile, and Leu) have the highest interaction energies among all amino acids in the PDB of (−4 to −6 RT units) (*Chan et al., 2004*). At the other end, mutations to small branched amino acids (Ser, Cys, and Ala), which have low interaction energies, were most depleted among dead variants (12%) vs. the

not-dead group (18%), suggesting that substitutions to small branched amino acids are least likely to inactivate allosteric signaling.

To understand at an atomic level why aromatic mutations are consistently enriched for in dead variants, we examined modeled phenylalanine and tyrosine inactivating mutations in TetR. These mutations were chosen as they were consistently among the most prevalent mutations in dead variants. In the inactive C144F TetR variant, F144 formed strong aromatic interactions at the dimer interface with its neighbor F140 and H139 in the other monomer, potentially stabilizing the inactive state of the protein through increased dimerization interactions (*Figure 3C*). Similarly, the inactive Y199 mutation may stabilize the inactive state by creating increased interactions and surface area between the monomers.

We concluded that the interaction energy of the large hydrophobic sidechains provides an enthalpic gain that stabilizes the allosteric OFF state of the protein. The resulting increase in energy gap, relative to wild-type, makes the variant unresponsive to ligand-induced allosteric activation. Thus, evolution of allosteric proteins is constrained to sequence variations that maintain an appropriate energy gap between ON and OFF states. Our results suggest that a few RT units can tip this delicate balance toward the inactive OFF state.

## Discriminative features of allosteric hotspots vary among homologous proteins

While the above analyses revealed several interesting features of hotspot residues, the partial overlap of these features between hotspot and non-hotspot residues suggests that additional features are required to make reliable predictions of allosteric hotspots. Prior to establishing such a predictive model, it is important to first understand what features are most likely to differentiate hotspot from non-hotspot residues. Accordingly, we assembled a comprehensive list of 27 features that are potentially relevant to the classification of a protein site as an allosteric hotspot (see Materials and methods) (*Figure 4A*). These include eight intrinsic physicochemical properties of amino acids such as charge and hydrophobicity, and eight local structural properties, such as solvent accessibility, local structural entropy (LSE) (*Jenik et al., 2012*), and frustration index (*Chakrabarty and Parekh, 2016*). Since allostery is fundamentally about cooperativity between distant sites in a protein, we also included 11 global features that describe long-range structural or dynamical properties; for example, residue centrality (*Bahar and Rader, 2005*), which measures the degree of connectedness of a residue when the protein is represented as a graph; a residue's distance to important regions in the protein, such as the DNA/ligand-binding sites and local peaks of residual centralities (see Materials and methods); motional covariance between a residue and the ligand/DNA-binding residues evaluated using an elastic network model (*Xia et al., 2010*).

To quantify how well these features differentiated hotspots and non-hotspots, we computed two metrics commonly used to estimate feature importance in machine learning – the F score and the Jensen-Shannon divergence (JSD). The F score measures the difference between the average (mean) of the two distributions relative to the widths of these distributions; the F score of feature i is computed as (*Pan et al., 2018*),

$$F_i = \frac{\left| \bar{x}_{hi} - \bar{x}_{ni} \right|}{\sigma_{hi} + \sigma_{ni}} \qquad (1)$$

in which $\bar{x}_{hi}$ / $\bar{x}_{ni}$ are the average values of hotspots/non-hotspots, and $\sigma_{hi}/\sigma_{ni}$ are the corresponding standard deviations. A large F score indicates that a specific feature adopts significantly different values for hotspot and non-hotspot residues. Features with larger F scores better differentiate hotspots from non-hotspots than those with smaller F scores. For all four aTFs, global features have the highest F scores and physicochemical features tend to have the lowest F scores. The confidence of hotspot assignment increases with increasing dynamic range because there is a clearer separation of dead vs. not-dead. Since RolR and TtgR have lower dynamic ranges, this may be a factor in their lower F scores.

Since the distributions of the various features are not necessarily mono-modal, we also evaluated feature importance using JSD, which is different from F score in that it measures the overall similarity between two distributions of arbitrary shape rather than only the difference between the averages. A similar trend in the rankings of features was observed with JSD (*Figure 4—figure supplements 1–5*). These striking differences among the three classes of features clearly show that a residue's relative

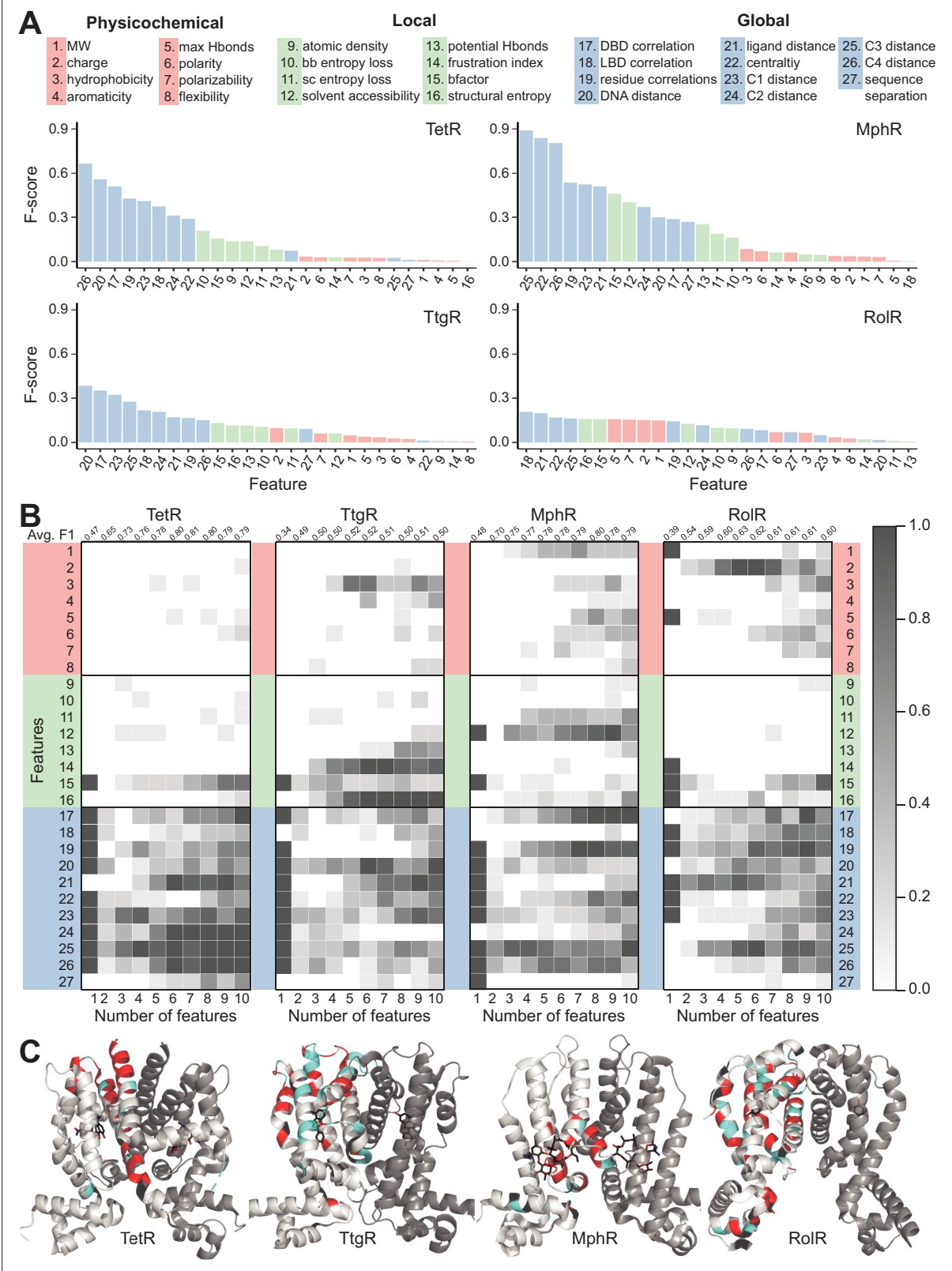

**Figure 4.** Machine learning identifies structural and molecular features that differentiate allosteric hotspots. (**A**) The full list of 27 features is shown at the top. The F scores (measure of importance) of the features for each of the four allosteric transcription factors (aTFs) is shown below. (**B**) Frequency of appearance of the 27 features in the top ten 1–10 feature combinations ranked by F1 score for each protein. Row 2–28 corresponds to feature 1–27, row 1 is the average F1 score of the top ten 1–10 feature combinations. (**C**) Predictions made by the model based on the best fivefold cross-validation

*Figure 4 continued on next page*

*Figure 4 continued*

performance achieved for each aTF (red: true positive; cyan: false positive; black: false negative; rest: true negative). The features used in the best models are 2, 19, 21, 23–26 for TetR; 4, 7, 10, 15, 19, 21, 24, 25 for MphR; 2, 10, 12, 21, 25 for RolR, and 9, 10, 13, 23, 25 for TtgR.

The online version of this article includes the following figure supplement(s) for figure 4:

**Figure supplement 1.** Global features have the highest Jensen-Shannon divergence (JSD).

**Figure supplement 2.** Distributions of TetR's hotspots' and non-hotspots' z-scored feature values for feature 1–27.

**Figure supplement 3.** Distributions of MphR's hotspots' and non-hotspots' z-scored feature values for feature 1–27.

**Figure supplement 4.** Distributions of TtgR's hotspots' and non-hotspots' z-scored feature values for feature 1–27.

**Figure supplement 5.** Distributions of RolR's hotspots' and non-hotspots' z-scored feature values for feature 1–27.

**Figure supplement 6.** Average and best F1 scores of 4–10 feature combinations converge after 10 generations in the genetic algorithm feature selection.

**Figure supplement 7.** Machine learning identifies structural and molecular features that differentiate allosteric hotspots.

**Figure supplement 8.** Positions of centrality peaks.

location in the protein structure and its motional correlation with other residues are more indicative of its role in allostery, as compared to its molecular features and local environment. In other words, once the protein fold is specified, the contribution of structure is larger than the contribution of sequence in determining the importance of a residue in allostery. On the other hand, substantial variations in the patterns of the F score and JSD among the four proteins highlight that the distinguishing properties of hotspot residues may differ even among homologous proteins, suggesting non-trivial variations in the detailed mechanism among them.

## NN analysis further highlights convergence and divergence in allostery mechanisms among homologous proteins

Having established the importance of individual features, we set out to build models by combining these features to reliably classify whether a protein site is an allosteric hotspot because combinations of features tend to perform better than models based on a single feature (*Wang et al., 2018*; *Ofran and Rost, 2007*; *Demerdash et al., 2009*; *Pethe et al., 2019*; *Gelman et al., 2021*; *So and Karplus, 1996a*). We included even the lower-ranked features (*Figure 4A*) as they may contribute to the discriminative power of a model when used in combination with other features in NNs. To search for the best feature combinations, we coupled the NNs with a genetic algorithm (GA), which has been shown to be efficient at picking out desired feature combinations when the total number of possible combinations is too large for an exhaustive search (*So and Karplus, 1996b*; *Halabi et al., 2009*). Specifically, we implemented the evolutionary programming algorithm to search for the best 1–10 feature combinations for NNs for each aTF. The algorithm guarantees that the average and highest fitness of the gene pool increases monotonically with evolutionary time (number of generations iterated), or remains constant upon convergence (see Materials and methods for details); these properties are essential for the convergence and proper termination of the feature optimization process. The fitness during GA optimization is evaluated as the average F1 score (distinct from the F score) of five times of fivefold cross-validation tests. The F1 score is defined in *Equations 2–4*, where TP, FP, TN, and FN represent true positive rate, false positive rate, true negative rate, and false negative rate, respectively.

$$\text{Recall} = \frac{\text{TP}}{\text{TP+FN}} \tag{2}$$

$$\text{Precision} = \frac{\text{TP}}{\text{TP+FP}} \tag{3}$$

$$\text{F1} = 2*\text{Recall}*\frac{\text{Precision}}{\text{Recall+Precision}} \tag{4}$$

Both the average and best fitness scores converge after 10 generations for 4–10 feature combinations (*Figure 4—figure supplement 6*). As the total numbers of all 1–3 feature combinations are moderate, they are evaluated exhaustively without using the genetic algorithm.

The F1 scores of the best model and that of a random model for each protein are: 0.83 and 0.19 for TetR, 0.82 and 0.16 for MphR, 0.64 and 0.26 for RolR, and 0.54 and 0.21 for TtgR; the F1 score of a random model is given by the fraction of residues being identified as allosteric hotspots in the

DMS experiments. The prediction results of the best models are further visualized by mapping the best fivefold cross-validation performance onto the crystal structures of the four proteins (*Figure 4C*). The results illustrate that all best models significantly outperform their corresponding random models, demonstrating the effectiveness of combining subsets of the 27 features in describing hotspots. However, the performances of the models are not uniform across the four proteins, with RolR and TtgR exhibiting significant numbers of false positive hotspots predicted by the GA-NN models (blue regions in *Figure 4C*). This is consistent with the above observation that the same set of features may have different F scores and JSDs for different proteins, which suggests that the four proteins, despite being homologous, likely feature different detailed allosteric mechanisms.

Next, to identify the key features for differentiating hotspots from non-hotspots, we examined the frequency of appearance of all features in the top 10 NN models (ranked by F1 score) using 1–10 features in each case (*Figure 4B* and *Figure 4—figure supplement 7*). For example, models containing four features for TetR are dominated by features 25, 23, 19, and 17 (see vertically at number of features = 4 for TetR, *Figure 4B*). The average F1 score of the top 10 models using n features reach its maximum at 0.81 when n=7 for TetR, 0.8 when n=8 for MphR, 0.63 when n=5 for RolR, and 0.52 when n=5 for TtgR (see top row of *Figure 4B* and *Figure 4—figure supplement 7*). These are optimal models as further increasing the number of features in the NN likely incurs the problem of overfitting. Examining the vertical trends (*Figure 4B*), we note that for a given number (n) of features, the top features do not always have overwhelmingly higher frequencies of appearance, suggesting a considerable degree of redundancy in the features that are able to differentiate hotspot residues. Nevertheless, the five most frequent features in the optimal models of the four proteins consist of 71.4% global features, 14.8% local features, and 14.8% physicochemical properties of wild-type amino acids, further supporting the notion that global features are more indicative of the role of a residue in allostery. For example, distances to centrality peaks are among the most frequent global features for all four aTFs; this observation is consistent with the expectation that a residue close to highly connected regions in a protein has a high chance of being a hotspot, since its mutations are likely to cause significant perturbations to protein structure and/or energetics.

A closer examination of the most frequent features in the optimal models reveals considerable variations. For example, distances to DNA and ligand are more important for TetR and RolR than for the other two proteins. Motional correlations with DNA-binding region and other residues are important in MphR, while their significance is modest in TetR and RolR, and very low in TtgR. Moreover, while the five most frequent features are all global in nature for TetR, the third most frequent feature is solvent accessibility (local feature) for MphR; the most frequent feature is charge (physicochemical feature) for RolR, and the top three most frequent features for TtgR are LSE (local feature), frustration index (local feature), and hydrophobicity (physicochemical feature) (*Figure 4B* and *Figure 4—figure supplement 7*). These observations highlight that being able to exploit the synergy between different classes of features is crucial for an NN model to achieve high performance, and the role of a residue in allostery is likely determined by a range of factors, with the weights of different factors being substantially different even among homologous proteins.

Therefore, compared to the F score analysis for individual features, the GA-NN analysis has revealed a more nuanced view of properties indicative of allosteric hotspots. To rationalize these observations, we note that all aTFs are structurally divided into LBDs and DBDs, thus allostery relies on both intra-domain properties and inter-domain couplings. While inter-domain couplings dictate the communication between allosteric and active sites, corresponding to a 'contact relayed signal' view of allostery (*Wang et al., 2020*; *Motlagh et al., 2014*); intra-domain properties affect allostery by shifting the populations of different thermodynamic states, as described by the classical MWC (Monod-Wyman-Changeux) model and its recent variations (*Marzen et al., 2013*; *Hilser et al., 2012*; *Luo et al., 2021*). Thus, an allosteric hotspot might contribute to one of these two types of properties or both (*Gandhi et al., 2008*). Features indicative of long-range motional correlations are likely discriminative for hotspots important to inter-domain coupling, while features reflecting spatial location of a site (e.g., distance to regions of high centrality) are likely more discriminative for hotspots important to intra-domain properties. Therefore, we speculate that if hotspots of an aTF are mostly important only to inter- or intra-domain properties, global features are highly effective in building high-performing models, like in the case of TetR and MphR. However, if most hotspot residues contribute to both inter- and intra-domain properties, a high level of cooperativity and epistasis is likely essential within the

aTF, making hotspot identification intrinsically harder, such as in the case of RolR and TtgR. For these latter cases, global features alone become less effective, and local or intrinsic features like charge, hydrophobicity, LSE, and frustration index, which represent finer description of the local interactions of a site, appear more frequently in the top-performing models.

## Transfer learning improves cross-protein predictions

Next, we explored the possibility of predicting hotspots on TetR homolog using models trained with data of other TetR homologs (cross-protein prediction [CPP]) with and without transfer learning (TL). For a given train-test combination (e.g., train on TetR and test on MphR), the prediction accuracy is recorded as its CPP performance. Then, the model is further trained with 10% of the data for the test protein and used to predict on the remaining 90% data, and the prediction result is recorded as its CPP_TL performance (see Materials and methods).

CPP_TL significantly outperforms their CPP counterparts as well as the corresponding best models trained with 10% data for the test protein only (*Figure 5A*). Using MphR as an example, the NN trained with data of TetR, RolR, and TtgR without TL gives the worst performance among the models tested (*Figure 5B*), while the model trained with 10% MphR data leads to rather low precision as well, which is not unexpected since it has only seen 10% data during training. By contrast, when the model trained with homologous proteins is further refined with 10% data of MphR, it yields a performance close to the optimal NN model trained specifically for MphR using the same set of features and all MphR data. These observations demonstrate that an NN trained for one protein, although not directly applicable to a homologous protein with modest sequence identity (the highest pairwise sequence identity is 19.2%, *Supplementary file 1*), can still learn useful information about allostery in the latter. It's also noted that NNs trained with data of three proteins in general outperform NNs trained with the data of only one of them in predicting on the fourth protein (*Figure 5A*). This result can be attributed to the expectation that NNs trained on the data of more proteins are less biased by the distinct characteristics of any single member, thus can learn common rules of allostery in all homologous proteins, leading to better prediction on a new protein unseen during training. In other words, while direct CPP performance is low due to the divergence in detailed allostery mechanism among homologous aTFs as discussed in the last subsection, the observation that TL can be effective suggests that such divergence can be represented by a limited amount of mutation data.

Only a fraction of the thousands of TetR-like proteins has three-dimensional (3D) structures. TL could be a powerful tool to predict hotspots in proteins with unknown structure but that have limited experimental data. To explore this idea, we repeated TL studies based on homology models rather than the crystal structure for all four aTFs (*Figure 6*). Specifically, we generated homology models using two different protein templates for each of the four proteins, and recalculated all structure-based features in each case. We observed a general trend of increasing model performance with increasing sequence similarity between template protein and MphR and decreasing model performance with increasing RMSD between template protein and MphR (*Figure 6* and *Figure 6—figure supplement 1*). Nevertheless, all relative performances are above 83%, which is remarkable considering that the lowest sequence identity between a modeled protein and its template is only 15.4% (*Supplementary file 4*), suggesting that the TL approach can be effective for predicting allostery in a homologous protein even in the absence of high-resolution structural information.

## Comparison with sequence-based featurization

In recent years, deep representation learning has emerged as an effective method for protein featurization. This approach, which generates a representation of a given sequence, is based on information extracted from the known protein sequence universe (*So and Karplus, 1996a*; *Biswas et al., 2021*; *Garruss et al., 2021*; *Freschlin et al., 2022*; *Alley et al., 2019*). In contrast, our approach is based on features derived from structural and physicochemical properties of amino acids. We sought to compare the performance of our model and a state-of-the-art sequence-based method, UniRep, which represents a protein sequence by a 1900-dimension vector (see Materials and methods for details) (*Werten et al., 2016*). Since UniRep features are based on protein sequences, an NN model trained with these features can be used to predict the protein phenotype upon every mutation (*Table 1*). For the prediction of hotspots (*Table 2*), we first rank all sites based on the fraction of predicted dead mutations for each site. The top N sites are then identified as hotspots, where N is the number of

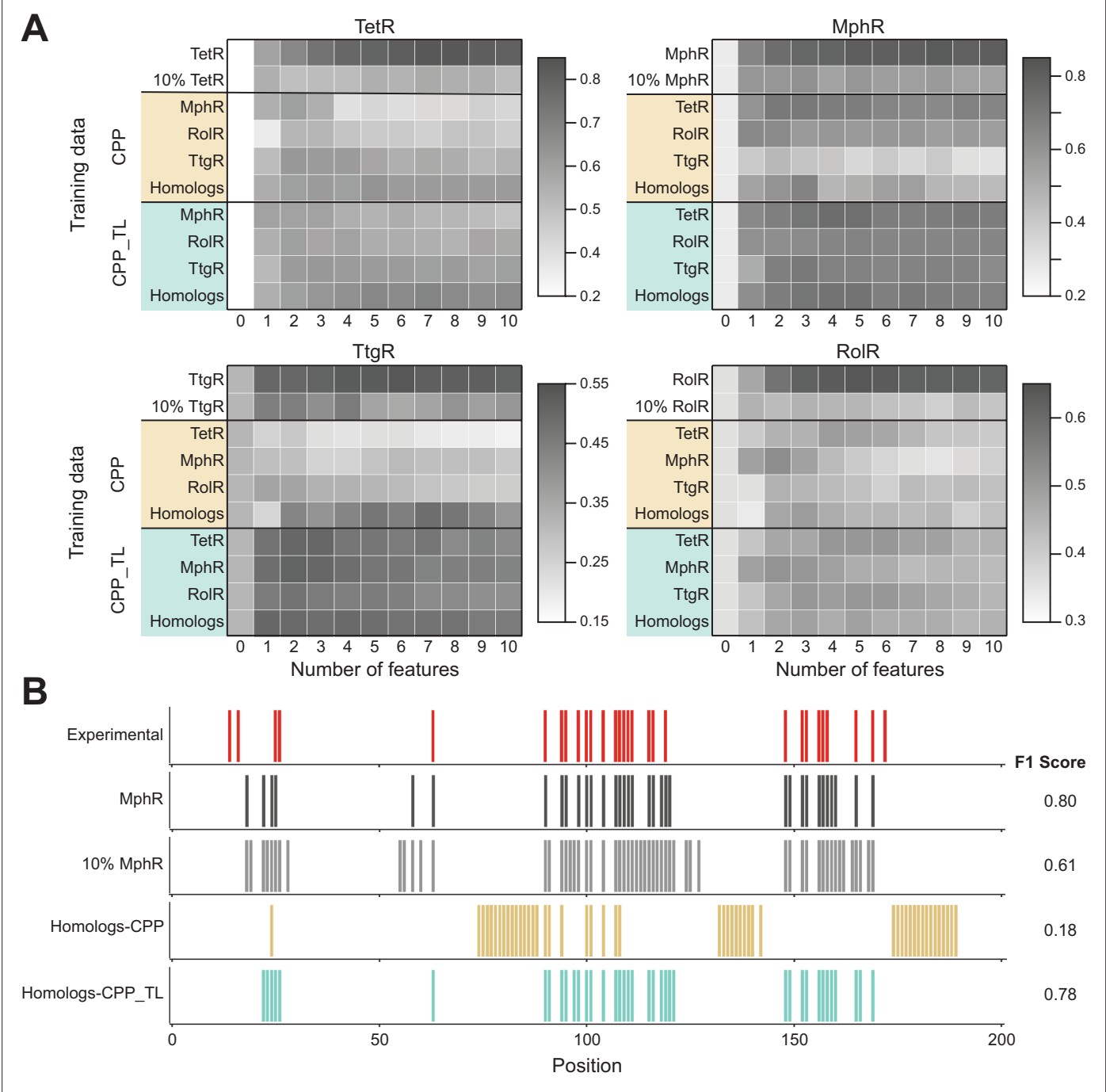

**Figure 5.** Cross-protein prediction – predicting allosteric hotspots in one homolog using data from other homologs. (**A**) Best cross-protein predictions without (CPP, yellow) and with transfer learning (CPP_TL, green) achieved for each protein using models trained with 1–10 features and different training data. The title of each heatmap specifies the target protein being predicted. The label of each row indicates the training dataset used (a protein name means data from that one protein and homologs means data from all other three proteins besides the target protein). The first row reports the best fivefold cross-validation performance achieved using 1–10 features on the target protein, and the first column (marked '0') is the performance of a random model for comparison. The row of 10%_TargetProtein indicates the best performance of neural networks (NNs) trained with only 10% data of the target protein in predicting the rest 90% data. (**B**) Comparison of hotspot predictions of MphR using different models (all employing features 17, 20, 21, 25, 26). Residue numbers of MphR are marked horizontally. Experimental data is the first row; 'MphR' shows the result of fivefold cross-validation performance of the NN on the MphR data; '10% MphR' shows the performance of NN trained with 10% MphR data in predicting the rest 90%; 'Homologs – CPP' and 'Homologs – CPP_TL' shows cross-protein prediction without and with transfer learning of NN trained with data of the other three homologs in predicting MphR.

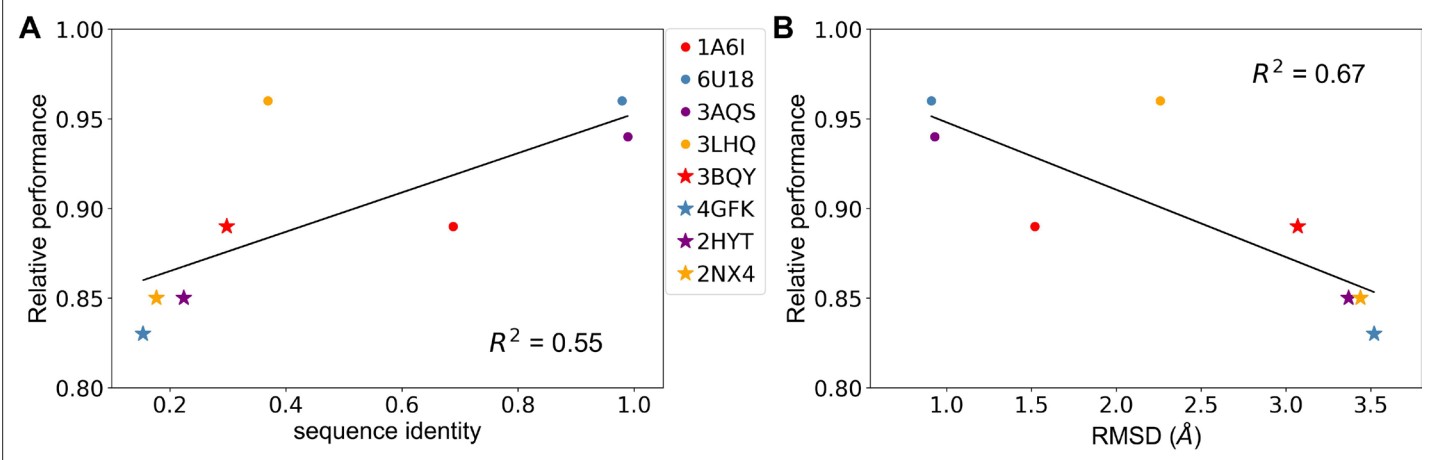

**Figure 6.** Predicting allosteric hotspots using homology models. (**A**) Correlation between relative performance and the identity between the template protein and the target protein for modeling. (**B**) Correlation between relative performance and the root mean squared distance (RMSD) between the template protein and the target protein for modeling. R squared shows the coefficient of determination of the corresponding linear regression (red: templates for TetR; blue: templates for MphR; purple: templates for RolR; orange: templates for TtgR).

The online version of this article includes the following figure supplement(s) for figure 6:

**Figure supplement 1.** Sequence identity and root mean squared distance (RMSD) between template protein and target protein are anticorrelated.

hotspots determined from DMS experiment for the protein of interest. We then evaluate the performance of the model by comparing the list of predicted hotspots with experimentally identified ones. When the UniRep features are combined with our 27 site features, the model can be used to predict mutation phenotypes and allosteric hotspots with the same procedure.

As summarized in *Tables 1–2*, UniRep feature-based models perform significantly better than the random baselines in both mutation phenotype prediction and hotspot prediction for all four homologous aTFs, highlighting the ability of such models in distilling fundamental features of a protein. However, there is a noticeable gap between hotspot prediction performance using UniRep features and our optimal models. This gap can be understood from the results of our above analysis using F score and JSD that global structure-based features are more indicative of allostery than sequence-based features. While our 27 features provide explicit and comprehensive descriptions of a residues' physicochemical property, location, motional correlation, and local environment in the context of the entire protein structure, sequence-based featurization attempts to infer such information from the sequence universe in order to establish an interpretable model without using structural information explicitly. When the 1900 UniRep features are combined with the 27 physical features in our model, a marginal yet consistent improvement in mutation phenotype prediction is observed (*Table 1*). This suggests that combining sequence-based features and features of clearer physical meaning can lead to improved predictive power. TL also proves to be effective in boosting the performance of cross-protein mutation prediction when the UniRep features and the 27 physical features are combined (*Supplementary file 5*).

**Table 1.** Mutation phenotype prediction performance[a].

|  | TetR | MphR | RolR | TtgR |
|---|---|---|---|---|
| UniRep1900 | 0.50±0.01 | 0.65±0.00 | 0.57±0.01 | 0.43±0.01 |
| feat1927 | 0.53±0.00 | 0.69±0.00 | 0.59±0.02 | 0.44±0.00 |
| random | 0.11 | 0.12 | 0.09 | 0.07 |

a. Performances are evaluated as the average performance of five times of fivefold cross-validation tests; Unirep1900 and feat1927 show best NN performance using only Unirep features and using Unirep features in combination with 27 physical features, respectively. Data are presented as average ± std.

**Table 2.** Hotspot prediction performance[a].

|  | TetR | MphR | RolR | TtgR |
|---|---|---|---|---|
| feat27 | 0.83±0.02 | 0.82±0.02 | 0.64±0.02 | 0.54±0.03 |
| UniRep1900 | 0.61±0.07 | 0.50±0.02 | 0.32±0.03 | 0.35±0.03 |
| random | 0.19 | 0.16 | 0.26 | 0.21 |

a. Feat27 represents the fitness of the best-performing feature combination emerged in feature selection with the GA-NN approach. Performances are evaluated as the average performance of five times of fivefold cross-validation tests, and presented as average ± std.

## Concluding remarks

Prediction of residues essential to protein allostery is of great fundamental and biomedical significance. An important question is to what degree allosteric hotspots and, therefore, mechanistic details of allostery, are conserved among homologous proteins. In this study, by combining DMS and machine learning analyses of four homologous aTFs, we have gained new understanding of this question. DMS has enabled a systemic, function-centric, approach to identify allosteric hotspots in proteins. Analysis of the distribution and basic properties of allosteric hotspots in the four aTFs has revealed key insights. First, hotspot residues are distributed across the structure rather than being limited to the specific pathway(s) that connect the inducer and DNA-binding sites as commonly assumed in allostery models. Nonetheless, they are relatively enriched near the dimerization interface and α4 helix at the LBD/DBD interface, highlighting the role of these regions in signal transmission (*Jumper et al., 2021*). Second, we observe that LRIs (in terms of sequence separation) are more prevalent among hotspots, suggesting that in addition to being important to folding and stability, LRIs are also relevant to propagating allostery signals. Third, a systematic analysis of F scores of a diverse set (*Fowler and Fields, 2014*) of protein site features suggests that in all four homologs, global structural and dynamic properties such as distance to centrality peaks, motion covariance with inducer/DNA-binding site residues are more useful than local and intrinsic physicochemical properties for differentiating hotspot from non-hotspot residues. The importance of global properties to the identification of hotspots is further confirmed by GA-NN models that optimize the combination of features to best classify whether a protein site is an allosteric hotspot. Fourth, combined with TL, the GA-NN model trained for one protein can lead to a reasonable prediction of hotspots in a homolog. Further, GA-NN models built using homology-modeled structures rather than actual crystal structure also perform well. These results support the idea that a generally similar allostery mechanism is at play in these homologous proteins.

On the other hand, we also observe a notable degree of divergence in allosteric hotspot distribution and the features that best define them among the homologous proteins. For example, hotspots in TetR and MphR are more concentrated in the C-terminal end of the protein, while the distributions are more even across the structure and sequence in TtgR and RolR. Related to this difference, the top F scores for different features are substantially lower in TtgR and RolR, highlighting that the hotspots in these two proteins are less distinctive (in terms of the features we have examined) than in TetR and MphR; similarly, the accuracy of the GA-NN model is generally less compelling for TtgR and RolR, when compared to TetR and MphR. Moreover, while global properties are most important to an NN model for the prediction of hotspots in TetR and MphR, local and intrinsic physical properties also contribute in TtgR and RolR. As discussed above, these differences might suggest a higher level of complexity in the allostery mechanism in TtgR and RolR, in which the hotspot residues may contribute to both intra-domain properties and inter-domain coupling. Therefore, allostery mechanisms in homologous proteins may differ substantially in fine details, an observation that has significant implication to the prediction of allostery in related proteins. Along this line, it is satisfying to observe that the TL approach can be rather effective for CPPs; the fact that the performance is not highly sensitive to the resolution of the structural model is particularly encouraging, especially considering recent advances in protein structure predictions (*Orth et al., 2000*).

Finally, we acknowledge that the accuracy of the GA-NN model, especially for CPP, which is most meaningful from the perspective of application, is not uniform. As mentioned above, the accuracy is generally less compelling for TtgR and RolR, which apparently feature more uniformly distributed

hotspot residues and therefore present a higher degree of challenge for both mechanistic understanding and hotspot prediction. To further improve the accuracy of prediction, it is likely worthwhile to include features that better encode the 3D structure with, for example, convolution NN models. Moreover, we have considered only one structural state for each protein, while structural variations among different functional states (e.g., inducer bound vs. DNA bound) are expected to be informative (*Jumper et al., 2021*; *Kosuri et al., 2013*). Finally, while we found limited value in including generic sequence-based features in the current work, it is possible that a more protein-family-specific set of sequence-based features can better contribute. Ultimately, we envision that by judiciously combining different types of experimental data and machine learning techniques in the framework of theoretical models (e.g., variations of the MWC model), we are able to not only predict but interpret the contribution of hotspot residues, which will pave the way for rational engineering of allostery to achieve desired biological function.

## Materials and methods

### Plasmid construction

We constructed a sensor plasmid with TtgR (Uniprot #Q88N29) and RolR (Uniprot #Q8NR95) cloned into a low-copy backbone (SC101 origin of replication) carrying spectinomycin resistance. The ttgR gene was driven by a variant of promoter apFAB61 and Bba_J61132 RBS while the apFAB50 promoter and BBa_J61119 RBS expressed rolR (*Terán et al., 2003*). On a second reporter plasmid, superfolder (sf) GFP was cloned into a high-copy backbone (ColE1 origin of replication) carrying kanamycin resistance. In the TtgR reporter, sfGFP was under the control of the native promoter driving ttgA expression (*Li et al., 2012*) modified to contain canonical −10 (5′-TATAAT-3′) and −35 (5′-TTGACA-3′) and the g10 RBS. sfGFP in the RolR reporter was driven by the lac operon promoter with rolO (*Rogers et al., 2015*) upstream of −35 and Bujard RBS. To control for plasmid copy number, red fluorescent protein (RFP) was constitutively expressed with the BBa_J23106 promoter and Plotkin RBS (*Terán et al., 2003*) in a divergent orientation to sfGFP. Plasmid construction for TetR(B) was previously described (*Leander et al., 2020*) and pJRK-H-mphR from the Church lab was obtained for MphR (*Magoč and Salzberg, 2011*).

### Library synthesis

Comprehensive single-mutant libraries of TetR, TtgR, MphR, and RolR were generated by replacing all wild-type residues to all other 19 canonical amino acids starting at position 2 (total mutant sequences – TetR: 3914; TtgR: 3971; MphR: 3667; RolR: 4332). Oligonucleotides encoding each single point mutation were synthesized as single-stranded Oligo Pools from Twist Bioscience and Agilent. Due to limitations in synthesis length, oligonucleotide pools were organized into six to seven subpools spanning the encoding region for each homolog and were encoded and amplified as previously described (*Leander et al., 2020*). Regions of the sensor plasmids corresponding to the oligonucleotide subpools were amplified with primers linearizing the backbone, adding a BsaI restriction site, and removing the wild-type sequence. Vector backbones were further digested with DpnI, BsaI, and Antarctic phosphatase before library assembly.

We assembled mutant sub-libraries by combining the linearized sensor backbone with each oligo subpool at a molar ratio of 1:5 using Golden Gate Assembly Kit (New England Biolabs; 37°C for 5 min and 60°C for 5 min, repeated ×30). Reactions were dialyzed with water on silica membranes (0.025 μm pores) for 1 hr before transformed into DH10B cells (New England Biolabs). Library sizes of at least 100,000 colony-forming units (CFU) were considered successful. MphR libraries were complete at this point. Cells (New England Biolabs) containing the reporter pColE1_sfGFP_RFP_kanR (DH5α for TetR and RolR, and DH10B for TtgR) were transformed with extracted plasmids to obtain libraries of at least 100,000 CFU.

### Fluorescence-activated cell sorting

Library cultures for each subpool were grown in triplicate for 16 hr at 37°C in lysogeny broth (LB) containing 50 μg/mL kanamycin and 100 μg/mL spectinomycin for TetR, TtgR, and RolR; MphR cultures were maintained with 100 μg/mL carbenicillin. Libraries were seeded from a 50 μL aliquot of glycerol stocks and grown to an $OD_{600}$ ~ 0.2 before being split in two and induced with 1 μM aTC, 500 μM Nar,

1 mM Ery, or 7.5 mM Res, and grown overnight. Saturated (un)induced sub-library cultures were split into two groups and pooled based location in the gene for sorting and sequencing: sub-libraries 1-3 covered the N terminus while sub-libraries 4-6/7 covered the C terminus of the homologs. Pooled sub-libraries were diluted 1:50 in ×1 phosphate buffered saline and fluorescence intensity was measured on an SH800S Cell Sorter (Sony). Remaining uninduced cultures were spun down and plasmids were extracted for next-generation sequencing to represent the presorted library, identifying all variants present in the library. For sorting, we first gated cells to remove debris and doublets and selected for variants constitutively expressing RFP; this gate was skipped for MphR which did not express RFP. The induction profile of each wild-type homolog was used as reference in drawing gates on GFP fluorescence (*Figure 1—figure supplement 1*). Uninduced and induced pooled sub-libraries were sorted between ~10 and 1000 RFU (based on fluorescence distribution of repressed, DNA-bound wild-type TetR homologs; *Figure 1—figure supplement 1*) to identify nonfluorescent, inactive variants. A total of 500,000 events were sorted for each gated population and cells were recovered in 5 mL of LB for 1 hr before antibiotics were added and cultures grown for an additional 6 hr until an $OD_{600} \sim 0.2$ was reached when cells were spun down and plasmids extracted for sequencing. Each library was grown, sorted, and sequenced in triplicate.

## NGS preparation and analysis

In total, three conditions were sequenced in triplicate for each homolog sub-library: (1) the presorted population, (2) the sorted nonfluorescent, uninduced population, and (3) the sorted nonfluorescent, induced population. Sub-libraries were prepared for sequencing with plasmids extracted from the each of the three populations, amplified with two primer sets in a two-step PCR, and sequenced using a 2×250 Illumina MiSeq run as previously described (*Leander et al., 2020*). Paired-end Illumina sequencing reads were merged with FLASH (Fast Length Adjustment of SHort reads) using the default software parameters (*Edgar and Flyvbjerg, 2015*). Phred quality scores were used to compute the total number of expected errors for each merged read (*Potter et al., 2018*). Reads exceeding the maximum expected error threshold of 1 were removed.

Before analysis, two separate normalizations were performed on the total sequence reads to compare and draw common thresholds (1) between experimental conditions and replicates and (2) across proteins. First, total sequencing reads were normalized to 200k total (100k for each sub-library) across all three conditions and replicates for each homolog. Next, reads were normalized to account for differences in theoretical size of each protein's single-mutant library. For example, reads of RolR (4332 possible mutants) increased by ×1.18 relative to MphR (3667 possible mutants) for a total of 236k reads. A read threshold of 5 was then applied across all replicates, conditions, and proteins to reduce sequencing noise; increasing this threshold to 10 reads did not significantly affect final analyses or positions identified as hotspots (*Figure 1—figure supplement 5*).

Variants that did not have at least 5 reads in all replicates of the presorted population were not considered present in the dataset (gray, *Figure 1—figure supplements 2–3*). To be classified as 'dead' within a single replicate, variants must have at least 5 reads in both the induced and uninduced sorted populations. There was good correlation between replicates in the number of dead variants identified a every position within the protein (*Supplementary file 2*). Dead variants were then given a score of 0, 1, or 2 based on how many replicates within a protein they were identified as dead in as a measure of confidence in calling these variants dead. These scores were then used to calculate a weighted score for every position in the protein based on the number and confidence of dead variants at that position using *Equation 5*:

$$\frac{(0*D1_x)+(1*D2_x)+(2*D3_x)}{Total_x} \tag{5}$$

At position x, D1 is the number of variants dead in one replicate, D2 is the number dead in two replicates, D3 is the number dead in three replicates, and Total is the total number of variants present in the dataset. Dead variants present in only one replicate were not considered confident enough to include in the weighted score and were discarded. The interquartile range of weighted scores for every protein was calculated and positions with weighted scores above the calculated Q3 were identified as allosteric hotspots (*Figure 1—figure supplement 4*). Sequencing data uploaded here. https://doi.org/10.5281/zenodo.7020077.

## Sequence conservation

Sequence conservation of TetR(B) was previously calculated (*Leander et al., 2020*). Homologs of TtgR, MphR, and RolR were identified using HMM search (https://www.ebi.ac.uk/Tools/hmmer/) (*Larkin et al., 2007*) against UniProtKB database with individual sequences used as queries. Sequences with alignment coverage less than 95% of full-length TtgR, MphR, and RolR were removed from consideration. The remaining sequences were aligned using Clustal Omega (*Waterhouse et al., 2009*). After applying a redundancy cutoff of 90%, we were left with 500–6000 which was used to evaluate sequence conservation score within Jalview (*Livingstone and Barton, 1993*). Conservation score in Jalview is computed by AMAS tool (*Vehlow et al., 2011*) and positions with a score of 7 or more were termed highly conserved. Two-sample t-tests were used to compare the average conservation score of residues classified as dead or no effect for each homolog. Ligand-contacting residues, defined as contacts within 5 Å of the ligand, were removed when calculating average conservation.

## Mapping, clustering, and ranking LRIs

Contact maps of TetR homologs were generated using CMView (*Kellogg et al., 2011*). Crystal structures of TetR (PDB ID: 4AC0), TtgR (PDB ID: 2UXU), MphR (PDB ID: 3FRQ), and RolR (PDB ID: 3AQT) dimers were obtained and removed from ligands and water molecules. Structures were edited to combine the two monomers and renumber residues to identify intermolecular dimer interactions. Interactions between α carbon atoms within 8 Å of were identified and a minimum sequence separation of 10 residues was set to select for LRIs. For each homolog, k-means clustering was used to identify subgroups of LRIs based on location similarity in the contact map. The elbow method was used to determine the optimal number of clusters in which the within-cluster sum of squares was minimized (*Figure 2—figure supplement 2*); 10 clusters were chosen for each homolog. Clusters within each contact map were then ranked based on the percent of unique hotspots within the cluster (*Supplementary file 3*). A paired t-test was used to compare the percentage of hotspot and non-hotspot residues within all four homologs participating in LRIs.

## Amino acid physiochemical properties

Physicochemical properties of mutations were compared by binning all substitutions that were dead or had no effect, removing ligand-contacting residues, across all four TetR homologs and calculating the average hydrophilicity, hydrophobicity, polarity, mass, SASA, and polarizability. A two-sample t-test was used to compare the means of the dead and no effect mutations for each of the six properties.

## ΔΔG calculations and structural models of mutations

The crystal structure of TetR(B) with bound [Minocycline:Mg]⁺ dimer structure was obtained and water molecules removed before calculations run; the bound ligand was also removed from TetR(B). All modeling calculations were performed using the Rosetta molecular modeling suite v3.9. Single-point mutants were generated using the standard ddg_monomer application (*Kawashima and Kanehisa, 2000*), which enables local conformational to minimize energy. Calculations were run at every position in protein for all 20 amino acids, generating 50 possible mutant and wild-type structural models for each protein variant. Structures with the lowest total energy from the 50 mutant and wild-type models were used to calculate ΔΔG and served as models for structural analysis.

## Calculation of physicochemical features (#1–8)

The eight physicochemical properties of wild-type amino acids, molecular weight, number of electrostatic charges, hydrophobicity, aromaticity, number of potential hydrogen bond, polarity, polarizability, and flexibility are obtained from the AAindex database (*Waterhouse et al., 2018*).

## Calculation of local structural features (#9–16)

The PDB structures or the modeled structures, generated using the SWISS-MODEL webserver (*Baxa et al., 2014*), of the four homologs were used for the calculation of any structure-based features in the corresponding cases. Local atomic density of a residue R was calculated as the number of atoms from other residues that are within 5 Å to any atom of the residue R. Backbone entropy loss and sidechain entropy loss of a residue, measuring the loss of conformational entropy of a residue upon protein folding, were calculated with the PLOPS webserver based on the crystal structure of the

protein (*Joosten et al., 2011*). SASA was calculated with the DSSP webserver (*Kabsch and Sander, 1983*; *Li et al., 2017*). The number of potential hydrogen bonds of a residue was calculated with the WHAT IF webserver maintained by the Vrient group at the Radboud University (The WHAT IF Web Interface (umcn.nl)). The single-residue frustration index was calculated with AWSEM-MD Frustratometer based on the crystal structure of the protein (*Chakrabarty and Parekh, 2016*). It measures how energetically favorable the wild-type residue is for its position in the 3D structure of the protein, when compared with the other 19 possible amino acid choices. X-ray crystallographic B-factor of a residue was obtained from the ENM webserver (*Li et al., 2017*). LSE measures the likelihood of the local sequence around the residue to change secondary structure. The calculation of LSE for a residue follows the method described by Hwang et al., which is based on the probability of the four 4-residue sequences that contain the target residue to assume eight different secondary structures as observed in the protein data bank (*Jenik et al., 2012*).

## Calculation of global structural features (#17–27)

The correlation of motion of a residue with the DNA or ligand region was calculated as the maximum absolute value of correlation it has with any of the 10 residues that are closest to DNA/ligand. Orientational cross-correlations between residue fluctuations are calculated using the ENM server (*Li et al., 2017*), the values vary from –1 (fully anticorrelated motions) to +1 (fully correlated). Maximum correlation of motion of a residue was calculated as the average of the five largest absolute values of correlation the residue has with any other residue. Distance of a residue to DNA was calculated by first modeling a DNA sequence of 15 nucleotide pairs to the proteins studied through structural alignment with the PDB structure of 1QPI. The distance between the α carbon of a residue and the closest DNA nucleotide, where the position of a nucleotide is represented by the position of its center of mass, was then calculated. Distance of a residue to ligand was calculated as the distance between the α carbon of the residue to the closest center of mass of a ligand.

Centrality scores of each residue were calculated using the Network Analysis of Protein Structures (NAPS) server (http://bioinf.iiit.ac.in/NAPS/) (*Bahar and Rader, 2005*). The unweighted atom pair contact network of each structure was generated using a 0–5 Å threshold and node centrality was measured by closeness, or the shortest distance of one position to all others in the network. Distances to centrality peaks were measured by first identifying four major peaks in *Figure 4—figure supplement 8* for each of the four homologs. Distances to peaks 1–4 for a residue are the distances between the α carbon of the residue to those of the four residues at the four peak positions (*Figure 4—figure supplement 8*).

Sequence propagation of a residue R was defined as the largest sequence separation between R and all other residues within 5 Å of R, as was measured by the distance between α carbon atoms.

## Machine learning methods

### Architecture of the NN

We used the Keras machine learning package to build and train fully connected feedforward NNs. All implemented NNs (except those involving UniRep features) have one hidden layer of 10 neurons (with RELU activation) and 2 neurons (with softmax activation) in the output layer. Xavier initialization, Adams optimizer, categorical cross-entropy loss function, and a learning rate of 0.0007 were used in all cases.

NNs using UniRep features alone (labeled as UniRep1900) and NNs using UniRep features in combination with the 27 physical features (labeled as feat1927) have the same architecture as the NNs mentioned above except for an additional batch normalization layer in front of the hidden layer of 10 neurons. Hyperparameters (learning rate, epochs, class weights in the loss function) are tuned in a grid search to maximize performance.

### Evaluation of different feature combinations for a given dataset

For a given dataset (e.g., data of a single protein), performance of different feature combinations is evaluated through fivefold cross-validation. Specifically, in fivefold cross-validation, the given dataset is randomly divided into five equal partitions. Each partition is used as the test set once, while the other four partitions are used as the training set for the NN. The fivefold cross-validation performance

is then evaluated as the average of the five test F1 scores. The fitness of a given feature combination is evaluated as the average performance of five times of fivefold cross-validation.

### Selection of best feature combinations

For a given dimension of the feature space (p), we used a genetic algorithm to select the best feature combinations in terms of their fitnesses for a dataset for the given p. Specifically, we start with a randomly generated initial gene pool (generation 1) containing 300 genes, with each gene being a different p-feature combination and its fitness evaluated and recorded. A point mutation (change of 1 feature) is then made to each gene (parent) to generate 300 new genes (sons) that have not been evaluated before. The sons are then evaluated, and the 300 fittest genes are selected from the composite pool (parents plus sons) to form the next generation.

### Cross-protein predictions

When making predictions on a test protein A using NNs trained with data of a different protein B, an NN is trained with all data of protein B using a certain feature combination, and make predictions on all data of protein A to obtain a test F1 score. Such CPP procedure is carried out for five times for a given feature combination, and its performance is evaluated as the five-time-average F1 score. All the top 300 p-feature combinations for protein B (the last generation obtained through the genetic algorithm optimization) are evaluated for CPP on other proteins, with p=1–10.

### CPPs with TL

Making CPPs with TL contains one more step than the above CPP procedure. Specifically, for a test protein A, its data is randomly partitioned into 10 equal subsets. For each subset, an NN trained with all data of protein B using a certain feature combination is further trained with one subset of data of protein A. The NN is then used to make predictions on the other nine subsets to obtain one test F1 score. The CPP (with TL) performance of the feature combination is then evaluated as the 10-time-average F1 score.

### Mutation phenotype and hotspot prediction using UniRep features

NNs using UniRep1900 features and feat1927 can be readily used for mutation phenotype prediction as UniRep generates a distinct 1900-dimensional vector to represent each mutant sequence. Feat1927 is generated by concatenating the 1900 UniRep features for a mutant and the 27 features of the mutation site. The performance of mutation phenotype predictions is evaluated as the average performance of five times of fivefold cross-validation tests.

The performance of hotspot prediction is calculated based on mutation phenotype prediction result. Specifically, in a fivefold cross-validation for hotspot prediction, all residues of a protein are divided into five random partitions, each serving as test residues once while the other four partitions serving as training residues. For each test residue, the model generates prediction for the phenotype of all of its mutations, with x percent of the mutations being predicted to be dead. Thus in a fivefold cross-validation test, the model generates an x value for every residue in the protein. Then, the top N residues (N is the number of true hotspots of a given repressor determined experimentally) with highest x values are identified as hotspots by the model, we then calculate the F1 score of the result by comparing with the list of true hotspots. The reported performance is the average F1 score of five times of such fivefold cross-validation test.

## Acknowledgements

This work is funded by NIH Director's New Innovator Award DP2GM132682 (SR) and Shaw Scientist Award (SR), NIH Molecular Biophysics Training Program T32 GM08293 (ML), and R35-GM141930 (QC). Development of the machine learning model was partially supported by grant ML-21–016 from the Dreyfus foundation (QC). Computational resources from the Extreme Science and Engineering Discovery Environment (XSEDE), which is supported by NSF grant number ACI-1548562, are greatly appreciated; part of the computational work was performed on the Shared Computing Cluster which is administered by Boston University's Research Computing Services (URL: https://www.bu.edu/tech/support/research/).

## Additional information

### Competing interests

Qiang Cui: Reviewing editor, *eLife*. The other authors declare that no competing interests exist.

### Funding

| Funder | Grant reference number | Author |
| --- | --- | --- |
| National Institutes of Health | DP2GM132682 | Srivatsan Raman |
| National Institutes of Health | R35GM141930 | Qiang Cui |
| National Institutes of Health | T32GM08293 | Megan Leander |
| The Camille and Henry Dreyfus Foundations, Inc | ML-21-016 | Qiang Cui |

The funders had no role in study design, data collection and interpretation, or the decision to submit the work for publication.

### Author contributions

Megan Leander, Conceptualization, Data curation, Formal analysis, Investigation, Visualization, Methodology, Writing - original draft, Writing - review and editing; Zhuang Liu, Conceptualization, Data curation, Software, Investigation, Visualization, Methodology, Writing - original draft, Writing - review and editing; Qiang Cui, Conceptualization, Resources, Software, Formal analysis, Supervision, Funding acquisition, Visualization, Writing - original draft, Project administration, Writing - review and editing; Srivatsan Raman, Conceptualization, Resources, Supervision, Funding acquisition, Investigation, Writing - original draft, Project administration, Writing - review and editing

### Author ORCIDs

Zhuang Liu ⓘ http://orcid.org/0000-0003-4695-7142
Qiang Cui ⓘ http://orcid.org/0000-0001-6214-5211
Srivatsan Raman ⓘ http://orcid.org/0000-0003-2461-1589

### Decision letter and Author response

Decision letter https://doi.org/10.7554/eLife.79932.sa1
Author response https://doi.org/10.7554/eLife.79932.sa2

## Additional files

### Supplementary files

• Supplementary file 1. Pairwise sequence identity and similarity.

• Supplementary file 2. R squared correlation of deads identified at each position between replicates.

• Supplementary file 3. Cluster rankings.

• Supplementary file 4. Template information.

• Supplementary file 5. Cross-protein prediction of mutation phenotype.

• MDAR checklist

### Data availability

Data included in the manuscript.

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
