## [Editor Report]

This article seeks to address a key question in protein biophysics: are the amino acid positions involved in allosteric mechanisms conserved across homologs of a protein family? Or do these mechanisms involve distinct amino acid patterns that vary amongst homologs? To address this question, the authors follow an innovative multidisciplinary approach that combines deep mutational scanning with machine learning; the findings of this study will be highly relevant to protein engineers and biophysicists.

---

## [Decision Letter]

**Decision letter after peer review:**

Thank you for submitting your article "Deep mutational scanning and machine learning reveal structural and molecular rules governing allosteric hotspots in homologous proteins" for consideration by *eLife*. Your article has been reviewed by 3 peer reviewers, and the evaluation has been overseen by José Faraldo-Gómez as the Senior Editor. All reviewers have opted to remain anonymous.

The reviewers have discussed their reviews with one another and with the Senior Editor, and the consensus is to invite you to submit a revised version of your manuscript that addresses the concerns enumerated below – particularly but not exclusively those put forward by Reviewer #3.

*Reviewer #1 (Recommendations for the authors):*

The paper is well written, the experiments are appropriately performed. I would like to encourage the authors to make the raw data available. I don't have any suggestions for changes to the manuscript and think that it makes a valuable contribution to the literature while noting that it leaves a few questions open-ended and that some of the speculation regarding the molecular basis could be tested experimentally. Obviously, this reflects my biases and interests as somebody interested also in structure and dynamics. Overall it's a very nice paper, congratulations.

*Reviewer #2 (Recommendations for the authors):*

In the Public Review I provided my overall comments. Below my aim is to make the concept and method clearer to the community.

With this aim in mind, there is one thing that I am missing. That is, how the authors define 'hotspots' via DMS. I think that it is important to clarify. This will help the readers.

I would also suggest to consider a figure comparing the hotspots in this paper to the hotspots as defined by the propagation pathways from the allosteric binding site to the distal active site. Would mapping these on a structure (e.g. using some server?) work? This could help the reader in visualizing the difference between the new concept suggested here and the 'traditional' definition.

*Reviewer #3 (Recommendations for the authors):*

Figure 1 supplement 1 sets the dynamic range of the assay and seems critical to the interpretation of the experiment. From these data, it would seem that the RolR assay lacks discriminating power across mutations. Consider promoting this figure to the main text.

We found the selection of the top 25% scoring residues as "allosteric hotspots" somewhat arbitrary – is it possible to instead select a cutoff based on the resolution (error) of the assay? Surely the top 25% of allosteric hotspots for RolR (which has a limited dynamic range) and TetR (which has a more extensive dynamic range) have very different biophysical effect sizes, and so this is not an apples-to-apples comparison?

Consider describing the behavior of previously well-characterized aTF mutations in your assay. This would help build confidence that the assay is truly reporting on allosterically dead mutations.

Consider using Fisher's exact test to assign a p-value describing the significance of enrichment/depletion of particular residue types in figure 3A

Figure 3C does not have a legend. Also, the figure seems to show K193Y while the main text refers to an inactive Y198 mutation.

In the methods section, it was unclear how the library was broken up. It seems like it was sequenced in two parts (to cover the entire open reading frame), but was this sequencing two regions of the same cultured and sorted sample? Or was the library broken into sublibraries that were cultured and sorted in smaller batches and then sequenced?

We understand that mutations at ligand-contacting positions were not considered, since these mutations are not allosteric (they instead directly affect ligand binding). How was ligand contacting defined?

Many times, plausible explanations/ideas seemed to be asserted as fact. We feel that these claims either need a citation, more expression of their uncertainty (explaining their lack of evidence in data and the literature), or a more careful explanation:

1. Abstract, page 2: "We found hotspots to be distributed protein-wide rather than being restricted to "pathways" linking allosteric and active sites as is commonly assumed" It is to our knowledge that it has never been asserted that allosteric hotspots themselves form pathways. Rather it is the assertion (in the prior literature cited by the authors) that allosteric hotspots preferentially contact coevolving networks of residues. In many cases, these co-evolving networks don't just link allosteric to the active site, but often connect other distal surfaces (with no known allosteric function) to the active site – going beyond the idea of a single pathway that the authors seem to imply.

2. Results page 5: "An aTF mutant that increases the thermodynamic gap between inactive and active states by stabilizing the inactive state will constitutively lock the protein in the inactive allosteric state. We term these "dead" variants. The dead variants are well-folded proteins that can bind to DNA and repress transcription but cannot be induced with ligand." The authors have made no measurements of protein stability. The only measurements made are cellular GFP levels in the presence and absence of the ligand. This needs a citation or some moderation of language.

3. Results page 6: "The evolution of aTFs has occurred through a series of gene duplication events resulting in mixing and matching LBDs and DBDs. " Citation needed.

4. Results page 6: "Thus, the DBDs likely exist as stand-alone domains that are not allosterically "wired" to the rest of the protein at the residue level but instead respond to large thermodynamic changes (e.g., inducer binding)." The data does not support this conclusion. The data suggests that the DBD is qualitatively depleted for mutations that abolish ligand-based activation while maintaining apo repression.

5. Results page 6: "Taken together, these observations show that although the hotspots are not superimposable across aTFs, the TetR-family likely share a conserved structural mechanism where the allosteric signal travels from the LBD through the dimer interface and a4 to the DBD, while the DBD itself acts as an internally rigid module that docks on DNA." This is speculation that seems more appropriate to the discussion (rather than results) section.

6. Results page 7: "These results also show that although allosteric hotspots may not be superimposable across distant homologs, local clusters of LRIs share similar patterns between homologs. As homologs get closer in sequence, regional similarities in allosteric signaling may give way to the superimposability of individual hotspots. " This again seems more appropriate to the discussion (rather than results) section.

7. Results page 8: "We concluded that the interaction energy of the large hydrophobic sidechains provides an enthalpic gain that stabilizes the allosteric OFF state of the protein." The thermodynamic mechanism of the mutations was not investigated in this study. We feel that it would be a stretch to form conclusions about enthalpy.

[Editors’ note: further revisions were suggested prior to acceptance, as described below.]

Thank you for resubmitting your work entitled "Deep mutational scanning and machine learning reveal structural and molecular rules governing allosteric hotspots in homologous proteins" for further consideration by *eLife*. Your revised article has been evaluated by the 3 original referees. I am glad to be able to inform the reviewers have decided to recommend that this work be published in *eLife*, pending revisions. As you will see below, one of the reviewers requires some additional clarifications in regard to the methodology, which might also help future readers to better appreciate the value of the work. Therefore we would like to offer you the opportunity to clarify these issues – which in my view would require editing of the manuscript and possibly moving Figure 1 S1 to the main text.

*Reviewer #1 (Recommendations for the authors):*

All the comments have been addressed satisfactorily in my opinion.

*Reviewer #2 (Recommendations for the authors):*

The revised manuscript addresses my comments/suggestions, and I think of the other reviewers as well. The paper is an excellent contribution to the literature in an important area and can be accepted as is. It is also an additional highly innovative and original work by the authors.

*Reviewer #3 (Recommendations for the authors):*

Thank you to the authors for a substantial revision. I very much appreciated the additions to the flow and NGS preparation/analysis methods sections, the clarifications on chip-based library construction, the more complete analysis of replicates, and the specification of the equation used to score allosterically dead variants. I also found the comparison to the ohm server predictions an interesting addition. As stated in my first review, the questions the authors seek to address – both (1) how allostery is implemented across homologs and (2)what physicochemical factors distinguish allosteric hotspots – are timely and fundamental open problems in protein biophysics. The paper contributes an enormous amount of experimental data, and the strategy of using machine learning to identify relevant features that distinguish hotspots is creative and leads to interesting results.

However, I still have substantial reservations about two aspects of the data analysis that persist from my earlier review. These are considerable enough that in parts of the manuscript I do not feel that the data support the authors' conclusions, but rather suggest something different. The first is that the strategy for deciding which mutations are allosterically dead still just doesn't make sense to me. I really might be missing something here; maybe the authors can explain. The second lies in the usage of quartiles to assign allosteric hotspots, and how this impacts the two constructs with a more limited allosteric dynamic range (RolR and TtgR).

Major concerns:

1. Strategy for assigning allosterically dead mutations. I appreciate that the authors clarified in their revisions that sequencing reads were normalized across replicates, experimental conditions, and proteins. These normalizations make sense to me, and I see that this helps in comparing counts. I also appreciate the authors' statements that "we use cell sorting as a binary classifier" and that by counting the number of dead mutations at a position they safeguard against noise in the data. Their point is well-taken that they seek to categorize mutations as allosterically dead/not dead, rather than using the flow data as a quantitative high-resolution measure. But I am still really stuck on understanding the use of a single threshold of 5 (or 10) to assess if a mutant is present in both the uninduced and induced sorted populations, and therefore assign it as allosterically dead. To illustrate, consider the following scenario. Let's imagine the authors ALSO sequenced the sorted induced fluorescent population (in addition to the induced nonfluorescent population). Now consider two different mutations with the following read distributions:

a. Mutant "A": 10 reads in the uninduced population, 10 reads in the induced non-fluorescent population, and 0 reads in the induced fluorescent population.

b. Mutant "B": 1000 reads in the uninduced population, 10 reads in the induced non-fluorescent population, and 900 reads in the induced fluorescent population.

If I understand correctly, both mutants would be classified as allosterically dead according to the authors' method. This makes sense for Mutant A, but for Mutant B…. it looks like it activates, just not completely, or maybe there is some noise in the sorting data. Is it obvious that if there are five or ten reads present it truly isn't noise? (How often do the authors observe "impossible codons" – meaning codons that are not part of their chip-based library – in the induced non-fluorescent population? This might set the noise threshold?) My impression is that the single threshold approach used by the authors may overestimate the number of "dead" mutations. It seems like it would be more correct to consider the ratio of the number of reads in the induced, sorted, non-fluorescent population relative to the number of reads in the uninduced population. One could then plot the distribution of this ratio (or maybe the log ratio), and apply a threshold to that ratio, rather than to threshold the absolute number of reads.

2. Strategy for assigning allosteric hotspots. Here the authors take the top quartile of residues according to their weighted positional score (that accounts for the number of dead mutations at a position as observed across replicates). By definition, this means that for each homolog, one-quarter of positions (however many were scored) will be called "hotspots". This seems consistent with what the authors report – for a length of 200 protein, you should then get about 50 hotspots. For RolR, which seems to be a bit longer, they get a few more (57 hotspots). So when they write that "changing the threshold (for assessing allosterically dead) has a modest impact on the overall number of hotspots" it is not really evidence of the robustness of the threshold choice – it is just that they are still taking the top quartile. If I understand correctly, they could use pretty much any strategy they like for assigning allosterically dead/not dead mutants and the number of hotspots would be about the same. That seems like an unusual feature of the analysis choice to me.

Now, the challenge is what happens when they consider TtgR and RolR. These are the two mutants with the least dynamic range in the assay (25-fold for TtgR, 15-fold for RolR, vs 49-fold for TetR, and 100-fold for MphR). When looking at the data in figure 1 supplement 2, it is clear that TtgR and RolR seem to have fewer allosterically-dead mutations per position. The matrices are overall less "stripey" in the vertical direction than TetR and MphR. So, when they take the top quartile of positions for TtgR and RolR to define allosteric hotspots, the cutoffs are much lower (~0.25-0.3) than for TetR and MphR (~0.8 or so, based on figure 1 supplement 3). As a consequence, what it means to be a hotspot in RolR or TtgR seems to be different than what it means in TetR and MphR. Indeed, my interpretation of the data (based on the heat maps in figure 1 supplement 3) would have been that RolR and TtgR just have fewer hotspots overall. This quartile-based definition of hotspots may explain a number of unusual features for RolR/TtgR, including the fact that: (1) the hotspot distributions are more diffuse across the sequence and structure (Figure 1) (2) that the F-scores are lower (they are less easily distinguished by physical properties), and (3) that the GA-NN model is less compelling. IMO, the reason that the GA-NN does less well for RolR/TtgR is that the training data is labeled improperly… basically, many of the positions they are calling hotspots are just not really hotspots. I feel like this is a far simpler explanation for differences in behavior for RolR and TtgR, rather than the authors' proposal that "… these differences might suggest a higher level of complexity in the allostery mechanism in TtgR and RolR, in which the hotspot resides may contribute to both intra-domain properties and inter-domain coupling". More generally, I think the choice of top-quartile means that what the authors compare across homologs is not truly apples-to-apples.

---

## [Author Response]

Reviewer #2 (Recommendations for the authors):Above, I provided my overall comments. Below my aim is to make the concept and method clearer to the community.With this aim in mind, there is one thing that I am missing. That is, how the authors define 'hotspots' via DMS. I think that it is important to clarify. This will help the readers.

Please see a detailed response to Reviewer 3 Question 11.

I would also suggest to consider a figure comparing the hotspots in this paper to the hotspots as defined by the propagation pathways from the allosteric binding site to the distal active site. Would mapping these on a structure (e.g. using some server?) work? This could help the reader in visualizing the difference between the new concept suggested here and the 'traditional' definition.

Thank you for the suggestion. In response to this suggestion, we predicted allosteric hotspots using Ohm webserver. The Ohm webserver is an efficient computational tool that analyzes the propagation of structural perturbation in proteins to identify allostery network and hotspot residues. The method is efficient and the resulting allostery network tends to be robust to small-scale variations in the input structure; it has been successfully applied to map allosteric networks in 20 systems for which high-resolution structures were available and the allosteric sites were known. The new supplementary figure (Figure 1 —figure supplement 7) compares allosteric hotspots between Ohm predictions and our experiments. The overlap between predictions and experiments is modest and involves mostly DNA binding domain residues while the experimental hotspots are distributed across the protein. This highlights the limitation of focusing on the mechanistic model that involves propagation of conformational distortions.

Changes to manuscript:

“We also compared the experimental hotspots with predictions made by the Ohm webserver(21). The Ohm webserver is an efficient computational tool that analyzes the propagation of structural perturbation in proteins to identify allostery network and hotspot residues. The overlap between predictions and experiments is modest and involves mostly DNA binding domain residues while the experimental hotspots are distributed across the protein (Figure 1 —figure supplement 7). This highlights the limitation of focusing on the mechanistic model that involves propagation of conformational distortions.”

Reviewer #3 (Recommendations for the authors):We found the selection of the top 25% scoring residues as "allosteric hotspots" somewhat arbitrary – is it possible to instead select a cutoff based on the resolution (error) of the assay? Surely the top 25% of allosteric hotspots for RolR (which has a limited dynamic range) and TetR (which has a more extensive dynamic range) have very different biophysical effect sizes, and so this is not an apples-to-apples comparison?

The reviewer makes an important point about classifying the top 25% scoring residues as allosteric hotspots being somewhat arbitrary. The most stringent and arguably unimpeachable definition of an allosteric hotspot is if every non-native mutation results in a dead phenotype (19/20 substitutions). However, few residues meet this strict condition because some mutations are inevitably partially active. This does not mean allosteric hotspots or “lynchpin” residues do not exist in a protein, only that any definition of an allosteric hotspot must be based on some arbitrary activity threshold.

Yes, the dynamic range of TetR and RolR are different. These differences are reflected in the distribution of weighted scores for each protein. In other words, the distribution of scores for each protein captures the biophysical effect size and the resolution of the assay. Our requirement that the dead mutation is considered only if it is present + and – inducer in both replicates applies to every protein. We do not alter this condition for RolR because it has a lower dynamic range. Therefore, comparing residue scores within a protein, not across proteins, is indeed “apples-to-apples.” We use this consistent definition to assign hotspots.

Consider describing the behavior of previously well-characterized aTF mutations in your assay. This would help build confidence that the assay is truly reporting on allosterically dead mutations.

Previous studies, which employ clonal screening, had identified a limited number of dead mutations (Hillen et al., PMID: 7552732). We found hundreds of dead mutations nearly seven times greater than what was previously known from clonal screens. Earlier studies identified hotspots in the region between LBD and DBD, though not comprehensively. But hotspots in other regions: short motif connecting α7 and α8, dimer interface on α8, and C-terminal end on α9, were previously known. We first reported them in our earlier study (PMID: 32999067) and now in this manuscript.

Consider using Fisher's exact test to assign a p-value describing the significance of enrichment/depletion of particular residue types in figure 3A

In figure 3A, we are merely observing trends in the enrichment and depletion of amino acids for comparison between the “dead” and “no effect” groups. We are careful not to make statistical claims regarding the significance of the enrichment/depletion. However, in figure 3B, we make statistical claims, backed by paired t-tests, on differences in physicochemical properties of both groups.

Figure 3C does not have a legend. Also, the figure seems to show K193Y while the main text refers to an inactive Y198 mutation.

We regret this error. The actual PDB residue number is 199. Our analysis software reassigns residue numbers when missing density is encountered, which changed the numbering to 198. We have changed the figure and main text to 199.

Changes to manuscript: Changed residue number to 199 in figure and main text.

In the methods section, it was unclear how the library was broken up. It seems like it was sequenced in two parts (to cover the entire open reading frame), but was this sequencing two regions of the same cultured and sorted sample? Or was the library broken into sublibraries that were cultured and sorted in smaller batches and then sequenced?

The full DMS library was broken into 6-7 sub-libraries spanning segments of the proteins. Each sub-library was cultured independently. They were then combined into two groups for sorting and sequencing – segments 1-3 in one group and segments 4-6/7 in the other.

Changes to manuscript: We added a few sentences under Materials and methods in the “Library synthesis” section.

We understand that mutations at ligand-contacting positions were not considered, since these mutations are not allosteric (they instead directly affect ligand binding). How was ligand contacting defined?

Residues with atom-atom contacts within 5A from the ligand in the crystal structure were considered ligand-binding.

Many times, plausible explanations/ideas seemed to be asserted as fact. We feel that these claims either need a citation, more expression of their uncertainty (explaining their lack of evidence in data and the literature), or a more careful explanation:1. Abstract, page 2: "We found hotspots to be distributed protein-wide rather than being restricted to "pathways" linking allosteric and active sites as is commonly assumed" It is to our knowledge that it has never been asserted that allosteric hotspots themselves form pathways. Rather it is the assertion (in the prior literature cited by the authors) that allosteric hotspots preferentially contact coevolving networks of residues. In many cases, these co-evolving networks don't just link allosteric to the active site, but often connect other distal surfaces (with no known allosteric function) to the active site – going beyond the idea of a single pathway that the authors seem to imply.

We agree with the reviewer that some literature, especially those from Ranganathan’s group, highlight the overlap between allosteric hotspots and coevolving residues to form “sectors” that connect protein surface, allosteric and active sites. Nevertheless, many recent publications on protein allostery continue to focus on specific pathways that link allosteric and active sites, while recognizing that multiple pathways may exist in a single system to form a network. The Ohm server (Ref. 21) discussed earlier in response to reviewer 2 question 7 is an example. Recent studies that integrate NMR relaxation measurements, MD simulations and network (community) analysis also tend to focus on such pathways (e.g., Ref. 17, 23). We have added reference to a recent study on allosteric pathways in the Cas9 protein using similar approaches (Ref. 24). Compared to these previous discussions, the hotspot distributions observed in our recent (Ref. 7) and current studies are much broader.

Changes to manuscript: New reference added to the following paper (Ref. 24).

“Enhanced specificity mutations perturb allosteric signaling in CRISPR-Cas9”, *eLife*, 2021

2. Results page 5: "An aTF mutant that increases the thermodynamic gap between inactive and active states by stabilizing the inactive state will constitutively lock the protein in the inactive allosteric state. We term these "dead" variants. The dead variants are well-folded proteins that can bind to DNA and repress transcription but cannot be induced with ligand." The authors have made no measurements of protein stability. The only measurements made are cellular GFP levels in the presence and absence of the ligand. This needs a citation or some moderation of language.

Changes to manuscript: We changed the main text as follows.

“We designate aTF mutations that constitutively lock the protein in an inactive allosteric state as “dead variants.” This may occur because the mutation stabilizes the inactive state by increasing the thermodynamic gap between inactive and active states. The dead variants are well-folded proteins that bind to DNA and repress transcription but cannot be induced with the ligand.”

3. Results page 6: "The evolution of aTFs has occurred through a series of gene duplication events resulting in mixing and matching LBDs and DBDs. " Citation needed.

Changes to manuscript: We have added citations to two publications, refs. 39 and 40. “Parallel evolution of ligand specificity between LacI/GalR family repressors and periplasmic binding proteins”, *Mol. Biol. Evol*., 2009

“Duplication of promiscuous transcription factor drives the emergence of a new regulatory network”, *Nature Communications*, 2014

4. Results page 6: "Thus, the DBDs likely exist as stand-alone domains that are not allosterically "wired" to the rest of the protein at the residue level but instead respond to large thermodynamic changes (e.g., inducer binding)." The data does not support this conclusion. The data suggests that the DBD is qualitatively depleted for mutations that abolish ligand-based activation while maintaining apo repression.

Changes to manuscript: We have removed “that are not allosterically wired to the rest of the protein at the residue level” from the manuscript.

[Editors’ note: further revisions were suggested prior to acceptance, as described below.]

Reviewer #3 (Recommendations for the authors):Major concerns:1. Strategy for assigning allosterically dead mutations. I appreciate that the authors clarified in their revisions that sequencing reads were normalized across replicates, experimental conditions, and proteins. These normalizations make sense to me, and I see that this helps in comparing counts. I also appreciate the authors' statements that "we use cell sorting as a binary classifier" and that by counting the number of dead mutations at a position they safeguard against noise in the data. Their point is well-taken that they seek to categorize mutations as allosterically dead/not dead, rather than using the flow data as a quantitative high-resolution measure. But I am still really stuck on understanding the use of a single threshold of 5 (or 10) to assess if a mutant is present in both the uninduced and induced sorted populations, and therefore assign it as allosterically dead. To illustrate, consider the following scenario. Let's imagine the authors ALSO sequenced the sorted induced fluorescent population (in addition to the induced nonfluorescent population). Now consider two different mutations with the following read distributions:a. Mutant "A": 10 reads in the uninduced population, 10 reads in the induced non-fluorescent population, and 0 reads in the induced fluorescent population.b. Mutant "B": 1000 reads in the uninduced population, 10 reads in the induced non-fluorescent population, and 900 reads in the induced fluorescent population.If I understand correctly, both mutants would be classified as allosterically dead according to the authors' method. This makes sense for Mutant A, but for Mutant B…. it looks like it activates, just not completely, or maybe there is some noise in the sorting data. Is it obvious that if there are five or ten reads present it truly isn't noise? (How often do the authors observe "impossible codons" – meaning codons that are not part of their chip-based library – in the induced non-fluorescent population? This might set the noise threshold?) My impression is that the single threshold approach used by the authors may overestimate the number of "dead" mutations. It seems like it would be more correct to consider the ratio of the number of reads in the induced, sorted, non-fluorescent population relative to the number of reads in the uninduced population. One could then plot the distribution of this ratio (or maybe the log ratio), and apply a threshold to that ratio, rather than to threshold the absolute number of reads.

We did not sequence the induced fluorescent population. Therefore, we cannot answer this question from data. However, there may be a flaw in the reviewer’s line of reasoning. The induced fluorescent population spans 4-6 orders of magnitude of fluorescence, depending on the protein. As a result, drawing the gate for the activity becomes difficult. To detect weak activity, as the reviewer points out, we would have drawn the gate adjacent to the non-fluorescent gate. This would dramatically increase false positives in the screen.

2. Strategy for assigning allosteric hotspots. Here the authors take the top quartile of residues according to their weighted positional score (that accounts for the number of dead mutations at a position as observed across replicates). By definition, this means that for each homolog, one-quarter of positions (however many were scored) will be called "hotspots". This seems consistent with what the authors report – for a length of 200 protein, you should then get about 50 hotspots. For RolR, which seems to be a bit longer, they get a few more (57 hotspots). So when they write that "changing the threshold (for assessing allosterically dead) has a modest impact on the overall number of hotspots" it is not really evidence of the robustness of the threshold choice – it is just that they are still taking the top quartile. If I understand correctly, they could use pretty much any strategy they like for assigning allosterically dead/not dead mutants and the number of hotspots would be about the same. That seems like an unusual feature of the analysis choice to me.

We may have a philosophical difference with Reviewer 3 on the concept of allosteric hotspots. We get the impression that Reviewer 3 considers the classification of a residue as a hotspot as a yes/no binary such that a residue is either a hotspot or not, and for a given protein, there is a fixed number of hotspots. While this view is not necessarily wrong, it ignores the notion of the magnitude of the contribution of a residue toward allostery, i.e., the ‘effect size’ of the residue. In contrast, we consider the effect size of a residue to fall along a spectrum – some residues are important lynchpins in allosteric signaling, whereas others may have an insignificant effect. Therefore, instead of binary classification, we first sought to rank the residues based on effect size. We then drew a cut-off for effect size on what we consider a hotspot based on the interquartile distribution of scores. In other words, though the hotspot classification is binary, the effect size is continuous.

To address the question that reviewer 3 posed on changing the read threshold not impacting the number of hotspots. The key takeaway from figure 1 —figure supplement 4 is that changing the read threshold does not change the identity of hotspots falling in the top quartile. This shows that varying the threshold does not impact the effect size of each residue. Our conclusions are, thus, robust to changing thresholds.

Changes to the manuscript: We have added the following sentence in the manuscript to clarify

“We note that changing the read threshold does not change the identity of hotspots falling in the top quartile indicating the robustness of our conclusions.”

Now, the challenge is what happens when they consider TtgR and RolR. These are the two mutants with the least dynamic range in the assay (25-fold for TtgR, 15-fold for RolR, vs 49-fold for TetR, and 100-fold for MphR). When looking at the data in figure 1 supplement 2, it is clear that TtgR and RolR seem to have fewer allosterically-dead mutations per position. The matrices are overall less "stripey" in the vertical direction than TetR and MphR. So, when they take the top quartile of positions for TtgR and RolR to define allosteric hotspots, the cutoffs are much lower (~0.25-0.3) than for TetR and MphR (~0.8 or so, based on figure 1 supplement 3). Indeed, my interpretation of the data (based on the heat maps in figure 1 supplement 3) would have been that RolR and TtgR just have fewer hotspots overall. This quartile-based definition of hotspots may explain a number of unusual features for RolR/TtgR, including the fact that: (1) the hotspot distributions are more diffuse across the sequence and structure (Figure 1) (2) that the F-scores are lower (they are less easily distinguished by physical properties), and (3) I feel like this is a far simpler explanation for differences in behavior for RolR and TtgR, rather than the authors' proposal that "… these differences might suggest a higher level of complexity in the allostery mechanism in TtgR and RolR, in which the hotspot resides may contribute to both intra-domain properties and inter-domain coupling". More generally, I think the choice of top-quartile means that what the authors compare across homologs is not truly apples-to-apples.

Yes, the dynamic range of the four aTFs is different. These differences are reflected in the distribution of weighted scores for each protein and captures the effect size of residues within that protein.

Let’s assume we could somehow measure the biophysical force associated with allostery disruption (e.g., conformational change) for a mutation. A protein with weaker allosteric activation may have a smaller force associated with a mutation than one with stronger allosteric activation. One can compare the force of disruption within a protein to create a ranked list of effect sizes of residues, but not across proteins. However, patterns of hotspots can be compared across proteins because each hotspot list is internally calibrated for that protein. In summary, the comparison of weighted scores within a protein is apples-to-apples, and the comparison of patterns (not scores) across proteins is also apples-to-apples.

As a consequence, what it means to be a hotspot in RolR or TtgR seems to be different than what it means in TetR and MphR.

Correct, and this is indeed the point. What it means to be a hotspot is a property of the protein, not a fixed threshold applicable to all proteins.

That the GA-NN model is less compelling. IMO, the reason that the GA-NN does less well for RolR/TtgR is that the training data is labeled improperly… basically, many of the positions they are calling hotspots are just not really hotspots.

We disagree with the reviewer that the training data is labeled improperly. The confidence of hotspot assignment increases with increasing dynamic range because there is a clearer separation of dead vs. not dead. Since RolR has lower dynamic range, this may be a factor in the lower F-scores in Figure 4A. This doesn’t mean the ‘data is improperly labeled’; it means the resolution of the assay is lower for RolR compared to TetR.

Changes to the manuscript: We have added the following sentence in the manuscript to clarify

“The confidence of hotspot assignment increases with increasing dynamic range because there is a clearer separation of dead vs. not dead. Since RolR and TtgR have lower dynamic ranges, this may be a factor in their lower F-scores.”